# Multi-Sensor Fusion for Wheel-Inertial-Visual Systems Using a Fuzzification-Assisted Iterated Error State Kalman Filter

**DOI:** 10.3390/s24237619

**Published:** 2024-11-28

**Authors:** Guohao Huang, Haibin Huang, Yaning Zhai, Guohao Tang, Ling Zhang, Xingyu Gao, Yang Huang, Guoping Ge

**Affiliations:** 1School of Mechanical and Electrical Engineering, Guilin University of Electronic Technology, Guilin 541004, China; botlowhao@126.com (G.H.); 13617824770@163.com (H.H.); zyn15512739923@163.com (Y.Z.); tgh1423065315@163.com (G.T.); 22012302137@mails.guet.edu.cn (L.Z.); 2School of Artificial Intelligence, Guangxi Minzu University, Nanning 541004, China; gaoxingyu@gxmzu.edu.cn; 3State Key Laboratory of Raw Silk and Silk Products Testing, Technical Center of Nanning Customs District, Nanning 530029, China

**Keywords:** multi-sensor fusion, wheel-inertial-visual odometry, iterative error state Kalman filter (IESKF), system noise covariance, fuzzy inference system (FIS)

## Abstract

This paper investigates the odometry drift problem in differential-drive indoor mobile robots and proposes a multi-sensor fusion approach utilizing a Fuzzy Inference System (FIS) within a Wheel-Inertial-Visual Odometry (WIVO) framework to optimize the 6-DoF localization of the robot in unstructured scenes. The structure and principles of the multi-sensor fusion system are developed, incorporating an Iterated Error State Kalman Filter (IESKF) for enhanced accuracy. An FIS is integrated with the IESKF to address the limitations of traditional fixed covariance matrices in process and observation noise, which fail to adapt effectively to complex kinematic characteristics and visual observation challenges such as varying lighting conditions and unstructured scenes in dynamic environments. The fusion filter gains in FIS-IESKF are adaptively adjusted for noise predictions, optimizing the rule parameters of the fuzzy inference process. Experimental results demonstrate that the proposed method effectively enhances the localization accuracy and system robustness of differential-drive indoor mobile robots in dynamically changing movements and environments.

## 1. Introduction

Position estimation in 6-DoF is important for determining the position of an autonomous navigation robot and its further actions in Simultaneous Localization and Mapping (SLAM) technology [1]. Proprioceptive and exteroceptive sensors are employed to accomplish localization and map-building tasks, which include obtaining external sensory information (i.e., the surrounding environment of visual light intensity or distance) and proprioceptive information (i.e., the robot’s internal values of speed and/or orientation) [2]. Proprioceptive sensors, such as wheel encoders, inertial measurement units (IMUs), cameras and LiDAR are mounted on the body of the robot to provide relative positioning information. In contrast, exteroceptive sensors including the Global Navigation Satellite System (GNSS) and Ultra-Wideband (UWB) systems are located outside the body of the robot to supply absolute position data.

The odometry system utilizes the relative position of each frame provided by proprioceptive sensors to estimate the instantaneous position and orientation of the robot relative to its starting point. This is a crucial module in SLAM and autonomous navigation technologies, especially for providing positioning and orientation data when exteroceptive sensors are unavailable. However, in the actual operation of indoor mobile robots, GNSS is limited in its ability to provide accurate indoor positioning due to signal blockages [3]. Although UWB enables localized positioning by deploying base station units to cover specific areas, expanding coverage requires additional base stations, which fails to meet the needs of large-area inspection tasks for indoor mobile robots [4]. Thus, using proprioceptive sensors in indoor environments is advantageous for real-time positioning of mobile robots. Nonetheless, it is generally recognized that single-sensor odometry performance is influenced by sensor characteristics, environmental disturbances, and vehicle state, often resulting in suboptimal localization accuracy. This limitation has led to extensive research on multi-sensor odometry approaches [5], such as Wheel-Inertial Odometry (WIO) [6], Visual-Inertial Odometry (VIO) [7], Wheel-Visual Odometry (WVO) [8], and LiDAR-Inertial Odometry (LIO) [9].

The focus of this study is a Wheel-Inertial-Visual Odometry (WIVO) with sensor fusion method to optimize the localization accuracy of a mobile robot. The wheel odometer (WO) calculates the rotation angle and linear displacement of a single wheel, the linear velocity of the left and right wheels, the linear velocity along the *x*-axis, and the angular velocity about the *z*-axis based on the number of pulses obtained by the wheel encoder, and finally provides translational and rotational motion information with six degrees of freedom through the Dead Reckoning method [10]. However, WO accuracy is affected by errors in rotational information and scale, especially under conditions like wheel slip, idling, wear, and uneven ground friction, impacting localization precision. Inertial Odometry (IO) employs a tri-axis gyroscope, accelerometer, and magnetometer. Since the accelerometer is susceptible to high-frequency noise (e.g., mechanical vibration and electromagnetic interference) [11] and the gyroscope to low-frequency noise (e.g., offset drift and random walk) [12], integrating acceleration and angular velocity over time provides translational and rotational information but lacks reliability. Visual Odometry (VO) offers small divergence in six-degree-of-freedom motion estimation between frames, but its sensitivity to environmental features can lead to global odometry divergence [13]. Therefore, the WIVO system proposed in this study includes two systems: Wheel-Inertial Odometry (WIO) and Wheel-Visual Odometry (WVO). The WIO subsystem estimates six degrees of freedom in movement using the linear velocity from wheel encoders and the angular velocity from the tri-axis gyroscope of IMU, addressing the rotational information errors in WO and the translational information errors in IO. The WVO subsystem combines WO information with two-frame VO, mitigating scale errors from wheel wear and uneven ground friction and reducing divergence in global VO. The odometry information from the WIO and WVO subsystems is integrated through sensor fusion to form the WIVO system.

To achieve more accurate robot localization, sensor fusion methods based on filtering algorithms are commonly used in back-end optimizations of SLAM multi-sensor fusion. Among these, Kalman filter variants are the most classic approach. The Kalman Filter (KF) was the first filtering method applied to SLAM sensor fusion, primarily based on the assumptions that the system is linear and that the noise follows a Gaussian distribution [14]. However, SLAM sensor fusion often requires filtering in nonlinear systems, which limits the application of the KF in SLAM. The Extended Kalman Filter (EKF) removes the assumption of linearity, using local linearization and Taylor series expansion to address the filtering challenges of nonlinear systems [15]. The Iterated Extended Kalman Filter (IEKF) is a modification of the EKF which adds an iterator for the observation step [16]. However, EKF and its variants linearize the measurement model using a Taylor series, which can cause significant errors in highly nonlinear systems, leading to biased state estimation in their performance [17]. The Unscented Kalman Filter (UKF) further optimizes the KF by avoiding the linearization process in the EKF and using the Unscented Transform (UT) to sidestep errors introduced by linearization [18]. However, the UKF suffers from arbitrary parameters necessary for sigma point placement, potentially causing it to perform poorly in nonlinear problems [19]. Besides, the Error State Kalman Filter (ESKF) is an indirect optimization approach compared to EKF, and it is particularly suitable for extended Kalman filtering on state spaces that lie on manifolds. By introducing error states, ESKF constrains the prediction and update steps to the manifold, allowing for more accurate handling of nonlinear problems [20]. The Iterated error-state Kalman Filter (IESKF) inherits the high speed from the Kalman filter, and the error state has smaller nonlinear errors than the original state during state estimation as with the ESKF. Additionally, it has been proved in [16,21] that its iterative update exhibits the same solving ability as the optimization method. With regards to the performance of IESKF, LINS [22] was the first method to employ the Iterated Error State Kalman Filter (IESKF) for tightly-coupled LiDAR and IMU-based robot motion estimation. It recursively corrected the robot’s estimated position using LiDAR-extracted features, preventing filter divergence during long-term operation while maintaining computational speed.

However, a fundamental issue remains in the previously mentioned Kalman filter-based fusion methods: the system noise covariance generally includes both the process noise covariance matrix and the observation noise covariance matrix [23], and the assumption that system noise covariance follows a Gaussian distribution with a mean of zero and constant variance is often unrealistic [24]. In general, methods for addressing this issue mainly fall into two categories: robust and adaptive approaches. Robust methods handle bounded uncertainties by solving a minimax problem to maintain stable performance under worst-case scenarios. However, they are often overly conservative, which compromises motion estimation accuracy and makes tuning difficult, such as the H∞ method [25] and Bayes KF method [26]. On the other hand, adaptive methods adjust algorithms in real time to adapt to changes, typically estimating the system noise covariance matrix based on innovation or residuals [27]. These methods rely on a sliding window that assumes consistency in errors, making them suitable only for slowly varying systems and inadequate for handling sudden disturbances and sensor failures. In addition to robust and adaptive approaches, the Fuzzy Inference System (FIS) has been extensively applied to adaptive control in complex systems with dynamic uncertainties [28]. It has been demonstrated to effectively establish nonlinear relationships between inputs and outputs with the support of human expertise and can be used in the field of dynamic prediction of system noise, which can quickly adapt to covariance prediction of robot motion and working environment in dynamic changes without mathematical models.

During the localization of a two-wheel differential speed mobile (TWDSM) robot, various complex kinematic challenges arise [29]. For WO, these challenges often occur in demanding scenarios, such as uneven wheel-to-ground friction or movements at significantly varying speeds during straight or turning motions. Under such conditions, the actual system noise does not conform to the predetermined process noise covariance matrix. For VO, similar issues are due to factors like varying lighting conditions and unstructured scenes (e.g., glass in the camera’s field of view) [30]. These factors dynamically affect the number of key points and reprojection error, rendering the actual observation noise incompatible with the predefined observation noise covariance matrix.

To address these problems, this paper proposes a multi-fusion method using a Fuzzy Inference System (FIS) to optimize the process and observe noise covariance in the Iterative Error State Kalman Filter (IESKF). The covariance matrices for process noise and observation noise are challenging to measure directly and are typically approximated through experience, system modeling and/or experimentation. However, the limitation of empirical estimation lies in its inability to effectively adapt to changes in system states or varying environmental scenarios. The proposed IESKF incorporates FIS and is built to adaptively adjust the position noise component of the covariance matrix in response to different motion states and environmental scenarios. The proposed method offers several advantages: (1) the IESKF filtering method applied in our paper not only inherits the high speed of the KF algorithm, but also utilizes error states and iterators to better solve nonlinear optimization problems; (2) incorporating FIS enables the system to quickly and dynamically adapt to covariance prediction of robot motion and working environment without relying on explicit mathematical models. The remainder of this paper offers the following:-The models of Wheel-Inertial Odometry (WIO) and Wheel-Visual Odometry (WVO) are formulated. Using the characteristics of WO and IO in translational and rotational motion, the WIO system is built. Additionally, the relatively stable interframe odometry information from VO and global translational odometry from WO are utilized to construct a WVO system. The formulated WIO and WVO models are then applied to the prediction and observation steps in IESKF, respectively.-An Iterative Error State Kalman Filter (IESKF) is applied based on WIO and WVO, ultimately forming a wheel-inertial-visual odometry (WIVO) multi-sensor fusion system. The key steps, including initialization, prediction, observation, error compensation and error reset, are demonstrated.-A Fuzzy Inference System is built based on WO (WO-FIS), focusing on the complex kinematic characteristics of a two-wheel differential robot under various motion scenarios such as forward movement, slow turning and high-speed spinning. By using the WO-estimated velocity differences, *z*-axis angular velocity, and *x*-axis linear velocity of the robot, an adaptive fuzzy inference is applied to the process noise covariance matrix in the prediction step of the IESKF, enabling a real-time adaptation of process noise covariance gain.-A Fuzzy Inference System is developed based on VO (VO-FIS), addressing conditions such as contrast reduction in input images due to varying lighting conditions and unstructured scenes (e.g., glass in the camera’s field of view). By using the information of visual keypoints and reprojection error, adaptive fuzzy inference is applied to the observed noise covariance matrix in the observation step of IESKF, enabling a real-time adaptation of observed noise covariance gain.

## 2. Multi-Sensor Fusion System with FIS-IESKF

The proposed multi-sensor fusion system consists of three main subsystems, as shown in Figure 1: (1) Odometry Fusion System, (2) IESKF System, and (3) Fuzzy Inference System. The Odometry Fusion System calculates WIO and WVO data based on inputs from the wheel encoder, IMU and RGBD camera. This odometry information is then used in the prediction and observation steps of the IESKF system, respectively. The IESKF system employs an IESKF to predict and/or update the robot’s state in real time by integrating fused odometry information into the robot’s kinematic and observation models. The Fuzzy Inference System includes the WO-FIS and VO-FIS, enabling real-time inference of robot motion information and visual feature points derived from the Odometry Fusion System. This facilitates dynamic prediction of the process noise covariance and observation noise covariance within the IESKF system, thereby meeting the position estimation requirements of high robustness under various complex motion scenarios and changing visual conditions.

### 2.1. Odometry Fusion System

The WIO system integrates linear velocity data provided by the wheel odometry and yaw angle change data from the inertial odometry as inputs. Using the Dead Reckoning method, these inputs are fused to compute translational motion information. Rotational motion information is derived by converting the pitch, roll, and yaw angles from the inertial odometry into quaternion orientation, ultimately forming the 6-DoFDoF odometry information for the WIO system.

The wheel odometry is calculated from the wheel encoder data. The fundamental principle of the wheel encoder is to measure wheel rotation to estimate linear displacement. It mainly consists of an encoder disk mounted on the axle and a fixed sensor. The encoder disk has evenly distributed marks or grooves, and as the wheel rotates, these marks generate pulse signals through the sensor [31]. The number of pulses is proportional to the wheel’s rotation angle *φ*, which can be calculated using Equation (1). Using the wheel rotation angle *φ* and the wheel radius *r*, the linear displacement *S* of each wheel can be calculated with Equation (2).
(1)$$\varphi =(M_{e}/N_{e})\cdot 360{}^{\circ}$$
(2)$$S=r\cdot \varphi \cdot \left(\pi /180{}^{\circ}\right)$$
where *M_e_* is the number of pulses received, and *N_e_* is the total number of marks on the encoder.

By integrating the linear displacement values *S_l_* and *S_r_* of the left and right wheels over time, the linear velocities *v_l_* and *v_r_* of each wheel can be calculated. Then, the linear velocity along the *x*-axis $v_{e}^{x}$ and the angular velocity around the *z*-axis $\omega_{e}^{z}$ of the robot’s center can be obtained through Equation (3).
(3)$$\left[\begin{array}{c} v_{e}^{x}\\ \omega_{e}^{z} \end{array}\right]=\left[\begin{array}{cc} 1/2 & 1/2\\ -1/d & 1/d \end{array}\right]\left[\begin{array}{c} v_{l}\\ v_{r} \end{array}\right]$$
where *d* is the axle length between the left and right wheels.

Inertial odometry information is derived from the fusion of data from the accelerometer, gyroscope, and magnetometer in a nine-axis IMU. Studies by Ojeda [32] and Harle [33] have demonstrated that IMU-based navigation systems can perform relatively accurate autonomous positioning and navigation in environments with buildings and tunnels, where GPS information is unavailable. These works highlight the independence and reliability of IMU-based systems in the absence of external signals. However, accelerometers are susceptible to high-frequency noise, meaning that directly double-integrating acceleration data can result in significant errors in linear displacement. Similarly, gyroscopes are affected by low-frequency noise, causing errors in rotational information if angular velocity data is integrated directly. To mitigate these issues and improve accuracy and stability in attitude estimation for devices like robots and drones, it is common practice to combine accelerometer, gyroscope and magnetometer data by the sensor-fusion method.

In this paper, we use a complementary filter [34] to fuse these sensor data for attitude estimation, obtaining the filtered rotation angles $\theta_{i}=[\theta_{i}^{x},\theta_{i}^{y},\theta_{i}^{z}]$ for (roll, pitch, yaw), respectively, as shown in Figure 2.

The rotation angle $\theta_{im}=[\theta_{im}^{x},\theta_{im}^{y},\theta_{im}^{z}]$ of the robot is derived from a combination of triaxial linear acceleration data $a_{i}=[a_{i}^{x},a_{i}^{y},a_{i}^{z}]$, magnetometer *B* = [*B_x_*, *B_y_*, *B_z_*], and gyroscope $w_{i}=[w_{i}^{x},w_{i}^{y},w_{i}^{z}]$. The rotation angles in the *x* and *y* directions are obtained using Equation (4). Since the magnetometer measures the strength of the Earth’s magnetic field along three orthogonal axes, the pitch and roll angles obtained in (4) allow for the determination of the magnetometer’s horizontal magnetic field *B_h_* and vertical magnetic field *B_v_*. Then, the yaw angle can be determined by Equations (5) and (6).
(4)$$\left\{\begin{array}{c} \theta_{im}^{x}=\arctan\left(a_{i}^{y}/a_{i}^{z}\right)\\ \theta_{im}^{y}=\arctan\left(-a_{i}^{x}/\sqrt{{\left(a_{i}^{y}\right)}^{2}+{\left(a_{i}^{z}\right)}^{2}}\right) \end{array}\right.$$
(5)$$\left[\begin{array}{c} B_{h}\\ B_{v} \end{array}\right]=\left[\begin{array}{ccc} \cos\left(\theta_{im}^{x}\right) & \sin\left(\theta_{im}^{x}\right)\sin\left(\theta_{im}^{y}\right) & \sin\left(\theta_{im}^{x}\right)\cos\left(\theta_{im}^{y}\right)\\ 0 & \cos\left(\theta_{im}^{y}\right) & -\sin\left(\theta_{im}^{y}\right) \end{array}\right]\left[\begin{array}{c} B_{x}\\ B_{y}\\ B_{z} \end{array}\right]$$
(6)$$\theta_{im}^{z}=\arctan(B_{v}/B_{h})$$

To mitigate the attitude estimation errors caused by high-frequency noise in accelerometer data and low-frequency noise in gyroscope data, a complementary filter is applied in this paper. This filter combines the low-frequency signals $\theta_{im}=[\theta_{im}^{x},\theta_{im}^{y},\theta_{im}^{z}]$ from the accelerometer and magnetometer with the high-frequency signals $w_{i}=[w_{i}^{x},w_{i}^{y},w_{i}^{z}]$ from the gyroscope, resulting in the filtered rotation angle information as shown in Equation (7).
(7)$$\theta_{i}=\lambda \cdot \left(\theta_{i}+\omega_{i}\cdot \Delta t_{i}\right)+\left(1-\lambda \right)\cdot \theta_{im}$$
where *λ* is the filter coefficient used to adjust the weighting between the gyroscope and the accelerometer/magnetometer for attitude estimation.

Subsequently, the *x*-axis linear velocity $v_{e}^{x}$ and the yaw angle $\theta_{i}^{z}$ from the inertial odometry can be used to solve the WIO of the TWDSM robot. The translational motion of the TWDSM robot is generally calculated using the Dead Reckoning method. The principle of this algorithm is to estimate the current position of translational motion by continuously integrating the displacement with the known linear velocity and yaw angle during movement [35].

As shown in Figure 2, *d* represents the wheelbase, {***W***} denotes the world coordinate system, and {***B_i_***} and {***B_j_***} are the robot base coordinate systems at times *i* and *j*, respectively. The *x*-axis linear velocity $v_{e}^{x}$ is obtained from the wheel encoder, while ${\Delta \theta }_{e}^{z}$ is the yaw angle change from time *i* to *j* as given in Equation (4). Similarly, ${\Delta \theta }_{i}^{z}$ is the yaw angle change obtained from the IMU through complementary filtering from time *i* to *j*. In general, when using the Dead Reckoning method for translational motion calculation, the TWDSM robot relies solely on wheel odometry data. As can be seen from Equations (1)–(4), this approach results in inaccuracies in the linear displacement *S* calculation when the robot encounters conditions like turning slippage or uneven ground friction, leading to inaccuracies in the real-time yaw angle calculation ${\Delta \theta }_{e}^{z}$ and restricting its effectiveness under actual ground conditions. When the computed linear displacement *S* does not accurately reflect the robot’s actual motion, additional errors may propagate into the *x*-axis velocity $v_{e}^{x}$ derived from the wheel encoder. However, compared to using the IMU triaxial acceleration data directly for second-order integration, which is more affected by high-frequency noise, the *x*-axis linear velocity $v_{e}^{x}$ obtained from the wheel encoder is more reliable. 

Therefore, the proposed method for constructing the translational motion WIO primarily uses $v_{e}^{x}$ as the *x*-axis linear velocity and ${\Delta \theta }_{i}^{z}$ as the yaw angle change in Equation (8).
(8)$$\left\{\begin{array}{c} p_{\mathrm{WIO}}^{x}=p_{\mathrm{WIO}}^{x}+v_{e}^{x}\cdot \cos\left(\Delta \theta_{i}^{z}\right)\cdot dt\\ p_{\mathrm{WIO}}^{y}=p_{\mathrm{WIO}}^{y}+v_{e}^{x}\cdot \sin\left(\Delta \theta_{i}^{z}\right)\cdot dt\\ p_{\mathrm{WIO}}^{z}=0 \end{array}\right.$$
where *P_WIO_* represents the position information from the WIO odometry translational motion, *dt* represents the differential of time *t*, and $p_{\mathrm{WIO}}^{z}$ is set to zero assuming the robot is moving on a flat indoor surface.

The rotational motion in WIO is obtained by converting the rotation angles through complementary filtering into quaternion orientation:(9)$$\left[\begin{array}{c} q_{\mathrm{WIO}}^{w}\\ q_{\mathrm{WIO}}^{x}\\ q_{\mathrm{WIO}}^{y}\\ q_{\mathrm{WIO}}^{z} \end{array}\right]=\left[\begin{array}{c} \cos\left(\frac{\theta_{\mathrm{WIO}}^{x}}{2}\right)\cos\left(\frac{\theta_{\mathrm{WIO}}^{y}}{2}\right)\cos\left(\frac{\theta_{\mathrm{WIO}}^{z}}{2}\right)+\sin\left(\frac{\theta_{\mathrm{WIO}}^{x}}{2}\right)\sin\left(\frac{\theta_{\mathrm{WIO}}^{y}}{2}\right)\sin\left(\frac{\theta_{\mathrm{WIO}}^{z}}{2}\right)\\ \sin\left(\frac{\theta_{\mathrm{WIO}}^{x}}{2}\right)\cos\left(\frac{\theta_{\mathrm{WIO}}^{y}}{2}\right)\cos\left(\frac{\theta_{\mathrm{WIO}}^{z}}{2}\right)-\cos\left(\frac{\theta_{\mathrm{WIO}}^{x}}{2}\right)\sin\left(\frac{\theta_{\mathrm{WIO}}^{y}}{2}\right)\sin\left(\frac{\theta_{\mathrm{WIO}}^{z}}{2}\right)\\ \cos\left(\frac{\theta_{\mathrm{WIO}}^{x}}{2}\right)\sin\left(\frac{\theta_{\mathrm{WIO}}^{y}}{2}\right)\cos\left(\frac{\theta_{\mathrm{WIO}}^{z}}{2}\right)+\sin\left(\frac{\theta_{\mathrm{WIO}}^{x}}{2}\right)\cos\left(\frac{\theta_{\mathrm{WIO}}^{y}}{2}\right)\sin\left(\frac{\theta_{\mathrm{WIO}}^{z}}{2}\right)\\ \cos\left(\frac{\theta_{\mathrm{WIO}}^{x}}{2}\right)\cos\left(\frac{\theta_{\mathrm{WIO}}^{y}}{2}\right)\sin\left(\frac{\theta_{\mathrm{WIO}}^{z}}{2}\right)-\sin\left(\frac{\theta_{\mathrm{WIO}}^{x}}{2}\right)\sin\left(\frac{\theta_{\mathrm{WIO}}^{y}}{2}\right)\cos\left(\frac{\theta_{\mathrm{WIO}}^{z}}{2}\right) \end{array}\right]$$
where *θ_WIO_* is the rotational motion and *q*_WIO_ is the quaternion orientation.

The linear acceleration {$a_{\mathrm{WIO}}^{k}$}*_k=x,y,z_* and angular acceleration {$\alpha_{\mathrm{WIO}}^{k}$}*_k=x,y,z_* for WIO need to be calculated using (10) by performing second-order differentiation on the translational and rotational motion information, respectively.
(10)$$\left[\begin{array}{c} a_{\mathrm{WIO}}^{k}\\ \alpha_{\mathrm{WIO}}^{k} \end{array}\right]=\left[\begin{array}{c} \frac{d^{2}}{dt^{2}}\left(p_{\mathrm{WIO}}^{k}\right)\\ \frac{d^{2}}{dt^{2}}\left(\theta_{\mathrm{WIO}}^{k}\right) \end{array}\right],\;k=x,y,z$$

In summary, the WIO in the proposed method can by constructed by (4)–(10). Since the accuracy of WO heavily depends on ground conditions, issues such as uneven ground friction or wheel wear can cause scale discrepancies and odometry drift over time. While VO does not rely on ground conditions, it can suffer from long-term drift due to lighting changes and unstructured scenes. To address the translational and rotational motion calculation errors in the wheel odometry and the IMU, a WVO is constructed in this paper. 

The construction of VO involves three main steps [36]: image feature detection and matching, camera position estimation, and visual odometry formation. The ORB feature is used for feature detection and matching which includes a FAST keypoints extractor and BRIEF descriptors. After detecting FAST keypoints, BRIEF descriptors and Hamming distance [37] are used to generate feature descriptors and match features accordingly. The method is applied for 2D feature points in the image coordinate frame. The EPnP method [38] is then used to estimate the position of a camera by solving the correspondence between known 3D points and 2D feature points, where the 3D points can be obtained from the depth values of the RGBD camera. Finally, the ICP method [39] is used to form the visual odometry, resulting in 3-DoFDoF translational information of the robot’s base coordinate {***B***} relative to the world coordinate {***W***} for VO. This is then fused with the 3-DoF wheel odometry to construct the WVO.

To construct the WVO, the first step is to establish the 3-DoF wheel odometry translational information. This is done by performing the Dead Reckoning method based on the robot’s linear velocity along the *x*-axis and angular velocity around the *z*-axis. 

The translational motion information of the WVO is obtained by linearly combining the WO and interframe VO, as shown in Equation (11). As the second step, the 3-DoF interframe VO is derived by calculating the difference between the visual odometry values of two consecutive frames {$p_{\mathrm{VO}}^{k}$}*_k=x,y,z_*, thus capturing the positional change between frames {$\Delta p_{\mathrm{VO}}^{k}$}*_k=x,y,z_*.
(11)$$\left\{\begin{array}{l} p_{\mathrm{WVO}}^{x}=\prime p_{\mathrm{WO}}^{x}+v_{e}^{x}\cdot \cos\left(\int \omega_{e}^{z}\cdot ds\right)\cdot dt+\Delta p_{\mathrm{VO}}^{x}\\ p_{\mathrm{WVO}}^{y}=\prime p_{\mathrm{WO}}^{y}+v_{e}^{x}\cdot \sin\left(\int \omega_{e}^{z}\cdot ds\right)\cdot dt+\Delta p_{\mathrm{VO}}^{y}\\ p_{\mathrm{WVO}}^{z}=0 \end{array}\right.$$
where {${\prime p}_{\mathrm{WO}}^{k}$}*_k=x,y,z_* represents the wheel odometry’s translational position along the *X*, *Y* and *Z* axes of the last frame, and {$p_{\;\mathrm{WVO}}^{\;k}$}*_k=x,y,z_* represents the translational motion information of the WVO of the current frame. The linear and angular velocity ($v_{e}^{x}$, $w_{e}^{z}$) along the *x*-axis and *z*-axis can be obtained by Equation (3). The term $\int \omega_{e}^{z}\cdot ds$ denotes the yaw angle calculated by integrating *z*-axis angular velocity over time. The second *dt* is to update the status of location for the WO. {${\Delta p}_{\;\mathrm{VO}}^{\;k}$}*_k=x,y,z_* is the inter-frame translational position information of VO, which is calculated by the subtraction value between two frames of {$p_{\;\mathrm{VO}}^{\;k}$}*_k=x,y,z_*. {$p_{\;\mathrm{VO}}^{\;k}$}*_k=x,y,z_* is constructed through image feature detection, camera position estimation and visual odometry formation. The position of WVO is computed by linearly superimposing the WO and VO position obtained from the above steps. The $p_{\mathrm{WVO}}^{z}$ component is set to zero under the assumption that the robot operates on a flat surface, where vertical movement (along the *z*-axis) is negligible. This simplification is valid for the specific application scenario addressed in this study and helps to improve the computational efficiency without impacting the overall accuracy of the system.

To minimize random noise from lighting disturbances, a sliding window averaging filter [40] is applied to the WVO. The concept of this filter is to compute the average of data within a moving window, thereby smoothing the signal and reducing noise.
(12)$${\hat{p}}_{\mathrm{WVO}}^{k}[n]=\frac{1}{M}\sum_{i=0}^{M-1}p_{\mathrm{WVO}}^{k}[n-i],\;k=x,y,z$$
where *M* represents the number of data points in each averaging calculation, *n* is the index of the current time or data point, and ${\hat{p}}_{\mathrm{WVO}}^{\;k}[n]$ is the filtered WVO value at time *n* constant. 

With the formulations in (11) and (12), the WVO is constructed. The 6-DoF position information obtained from both WIO and WVO will be applied to the prediction and observation stages of the IESKF system, respectively, to achieve multi-sensor data fusion and enhance the accuracy and robustness of position estimation.

### 2.2. Iterative Error State Kalman Filter (IESKF) System

In this paper, the IESKF algorithm is divided into initialization, prediction, observation, error compensation and error reset steps. Compared to the EKF (Extend Kalman Filter), which directly estimates nominal states, the IESKF iteratively linearizes and updates the error state during the observation process, achieving a higher level of linearity that better approximates the system’s true state [41]. As shown in Table 1, the true status **x*_t_*** represents the actual status of the system. The nominal status **x** is the estimate of the true status. The error status δ**x** denotes the deviation between the nominal status and the true status, defined as δx = **x*_t_*** − **x**. The error state is used to iteratively correct the nominal state, bringing it closer to the true state and thereby enhancing the accuracy of the estimation of the filter. In addition, the definition of the following 3-DoF state variables is given in this paper: position, velocity, rotation angle, linear acceleration bias, angular acceleration bias, and gravitational acceleration in terms of true state, nominal state and error state.

In Table 1, the rotation angle **θ** in the nominal state is derived from the quaternion-based attitude q_WIO_ of the WIO. The rotation angle **θ** and rotation matrix **R** are conversed through Euler angles in the IESKF, reflected in various state types such as nominal state, true state, and error state.

Figure 3 illustrates the flowchart of the IESKF system. The proposed IESKF has an added iterator for the observation step compared to the ESKF, as shown in Figure 3, where *N*[*i*] represents the current number of iterations, *N*_max_ is the maximum number of iterations and *ε* is the minimum value of iterations.

In the IESKF initialization, position from the WO is retrieved and set as the initial true position variable **p***_t_* with an approximation of initial state covariance matrix defined as $\hat{P_{t}}$.
(13)$$p_{t}=\left[\begin{array}{c} p_{t}^{x}\\ p_{t}^{y}\\ p_{t}^{z} \end{array}\right]=\left[\begin{array}{c} p_{\mathrm{WO}}^{x}\\ p_{\mathrm{WO}}^{y}\\ p_{\mathrm{WO}}^{z} \end{array}\right]\;\mathrm{and}\;{\hat{P}}_{t}=\mathrm{diag}\left({\hat{\sigma }}_{p}^{2}I_{3\times 3},{\hat{\sigma }}_{v}^{2}I_{3\times 3},{\hat{\sigma }}_{\theta }^{2}I_{3\times 3},{\hat{\sigma }}_{ba}^{2}I_{3\times 3},{\hat{\sigma }}_{b\alpha }^{2}I_{3\times 3},{\hat{\sigma }}_{g}^{2}I_{3\times 3}\right)$$
where ${\hat{\sigma }}_{P}^{2},\;{\hat{\sigma }}_{v}^{2},{\hat{\sigma }}_{\theta }^{2},\;{\hat{\sigma }}_{ba}^{2},\;{\hat{\sigma }}_{b\alpha }^{2},\;{\hat{\sigma }}_{g}^{2}$ denotes the initial noise variances for position, velocity, rotation angle, linear acceleration bias, angular acceleration bias and gravitational acceleration, respectively. **I**_3×3_ represents the 3 × 3 identity matrix.

In the IESKF prediction step, the WIO data is incorporated in the nominal kinematic equations of the TWDSM robot in (14), which is then transformed into the error-state kinematic Equation (15). These can then be converted to a state-space form as (16) and (17) for the prior estimate valuable δ**x^−^** and an approximation of prior estimate covariance matrix Pt−.
(14)fx18×1,u→pt+Δt=pt+vΔt+12Ra−baΔt2+12gΔt2vt+Δt=vt+Ra−baΔt+gΔtRt+Δt=RtExpα−bαΔtbat+Δt=batbαt+Δt=bαtgt+Δt=gt
(15)fδx18×1,u,w→δpt+Δt=δp+δvΔtδvt+Δt=δv+−Ra−ba∧δθ−Rδba+δgΔt+ηvδθt+Δt=Exp−α−bαΔtδθ−δbαΔt−ηθδbat+Δt=δba+ηaδbαt+Δt=δbα+ηαδgt+Δt=δg
(16)δx−=Fxδx
(17)Pt−=FxPtFxT+FiQFiT
(18)where δx=fδx,u,w=fδx,u+w,w~N0,Q,
(19)Q=diag03,Covηv,Covηθ,Covηa,Covηα,03=diag03,σ^v2I3×3,σ^θ2I3×3,σ^ba2I3×3,σ^bα2I3×3,03∈ℝ12×12
(20)Fx=∂f∂δxx,u=IIΔt00000I−Ra−ba∧Δt−RΔt0IΔt00Exp−α−bαΔt0−IΔt0000I000000I000000I∈ℝ18×18
(21)Fi=∂f∂ix,u=0000I0000I0000I0000I0000∈ℝ18×12

In (14)–(21), *u* = (**a**, **b_a_**, **a**, **b_α_**) is the control input values from WIO data including acceleration **a** and angular velocity **α** along with their respective biases **b_a_** and **b_α_**. **w** denotes the process noise, assumed to follow a normal distribution with a mean of 0 and an approximation of process noise covariance matrix **Q** as shown in Equation (18). **Q** is calculated as a diagonal matrix composed of various noise components (**η_v_**, **η_θ_**, **η_a_**, **η_α_**) shown in Equation (19). (**η_v_**, **η_θ_**, **η_a_**, **η_α_**) represent velocity, rotation angle, linear acceleration and angular acceleration noise, respectively. Δ*t* is the time interval. **F_x_** is the error-state transition matrix, and **F_i_** is the process noise control matrix.

In the IESKF observation step, the observation **z** refers to the data obtained from sensors or measurement devices. It represents a measurement of the system state and is used to update the filter’s estimate. The observation **z** is related to the system state **x** through the measurement model *h*(**x**), typically expressed as Equations (22) and (23) below.
(22)z=hx+v,v~N0,V
(23)V=diagσp2⋅I3×3∈ℝ3×3
where **z** = [pWVO, xpWVO, ypWVO z] represents WVO observations, **v** is the observation noise with an approximation of observed noise covariance matrix **V**, and **σ_p_**^2^ is the variance value of **V**.

In a traditional EKF, the observation equation is linearized with respect to the nominal state, obtaining the Jacobian for Kalman filter updates. In the IESKF, both the nominal status **x** and the error status δ**x** are used. Thus, the Jacobian of the observation equation **H** with respect to the error state is computed in (24), which executed the linearization operation for *h*(**x**). It is worth noting that the linearization operation may cause certain errors, therefore Kalman gain was calculated in Equation (25) and used to execute the posterior estimate in Equation (26) with the residual value (**z** − **H·**δ**x^−^**). Finally, the posterior estimated covariance was calculated in Equation (27).
(24)$$H=\frac{\partial h}{\partial \delta x}=\frac{\partial h}{\partial x}\cdot \frac{\partial x}{\partial \delta x}$$
(25)$$K=\frac{P_{t}^{-}\cdot H^{T}}{H\cdot P_{t}^{-}\cdot H^{T}+V}$$
(26)$$\delta x=K(z-H\cdot \delta x^{-})$$
(27)$$P_{t}=(I-K\cdot H)P_{t}^{-}$$
where K is the Kalman gain, δ**x** is the posterior estimate variable, and **P***_t_* is an approximation of posterior estimate covariance matrix.

After the prediction and observation steps, the error status δ**x** is built. Due to **x** being estimated by the nominal kinematic equations shown in (14), the superimposition between **x** and δ**x** is performed to derive the true status **x***_t_* by (28). It should be noted that for linear states (e.g., position, velocity), direct linear addition is sufficient. However, for nonlinear states (e.g., rotation, attitude), specific nonlinear operations, such as quaternion multiplication or Lie algebra mapping, are required. Finally, the error status variable is reset as δ**x** = **0**_18×1_.
(28)$$x_{t}=x⊕\delta x\Rightarrow \left\{\begin{array}{l} p_{t}=p+\delta p\\ v_{t}=v+\delta v\\ R_{t}=Re^{\delta \theta }\\ b_{a_{t}}=b_{a}+\delta b_{\alpha }\\ b_{\alpha_{t}}=b_{\alpha }+\delta b_{\alpha }\\ g_{t}=g+\delta g \end{array}\right.$$

### 2.3. Fuzzy Inference System (FIS) 

In the prediction and observation steps of the IESKF, both the process noise covariance and observation noise covariance are set as Gaussian distributions with a mean of zero and a fixed variance. However, this fixed-noise assumption cannot effectively adapt to the complex kinematic characteristics of the robot or the susceptibility of the visual observation model to variations in lighting conditions and unstructured scenes in the camera model. To address this problem, an FIS [42] is introduced to separately construct fuzzy models based on WO and VO, enabling real-time prediction of process noise and observation noise covariance values. 

The construction of the FIS involves five steps: determine fuzzy sets and their physical meanings, fuzzification, setting fuzzy rules, aggregation, and defuzzification. The proposed FIS in this paper is summarized in Table 2 and Figure 4 to show the relationship between fuzzy input/output variables and the operational performance of the robot for the first step.

In Table 2, each variable is used within the proposed FIS to interpret different operating conditions, such as slippage during turning, straight-line motion, or spin turning. These conditions are set by relevant fuzzy rules that adaptively adjust noise covariance values based on the current kinematic state and visual features of the robot. The *ρ* in Table 2 is defined as the wheel speed difference between the right and left wheels in a TWDSM. $v_{e}^{x}$, $\omega_{e}^{z}\;$ represents the linear velocity along the *x*-axis and *z*-axis, respectively. *N_u_* denotes the number of ORB feature points detected in the VO. *E_r_* is the ORB feature reprojection error, indicating the consistency of feature tracking.

As shown in Figure 4, the physical interpretation of the wheel odometry-based fuzzy inference system (WO-FIS) and Visual Odometry Fuzzy Inference System (VO-FIS) includes different operational effects:

(a)Left/right turn slippage (A1, A3): In Figure 4a, slippage during left or right turns is represented by the wheel speed difference *ρ*. When *ρ*∈A1, the linear speed of the right wheel significantly exceeds that of the left wheel. This speed difference implies that the right wheel surpasses its friction limit with the ground, causing the right wheel to lose traction and slip. Conversely, when *ρ* ∈ A3, slippage occurs on the left wheel as the left wheel speed exceeds the friction threshold.(b)Forward motion (A2): As shown in Figure 4b, during forward motion *ρ* ∈ A2, the left and right wheel speeds are equal, resulting in no wheel slippage.(c)Slow/Quick turn (A4, A5): In Figure 4c, a small angular velocity $w_{e}^{z}$ (small) around the *z*-axis represents a slow turn, where as a larger $w_{e}^{z}$ (large) indicates a fast turn.(d)Stationary/Spin-in-place (A6): Figure 4d demonstrates the robot’s stationary or spin-in-place conditions. When $v_{e}^{x}$ (0) and both wheel speeds are zero, the robot is stationary. When $v_{e}^{x}$ ∈ A6[1] but the left and right wheels have equal and opposite speeds $v_{e}^{x}$ ∈ A6[2], the robot rotates in place with an angular velocity $w_{e}^{z}$. However, if $v_{e}^{x}$ ∈ A6[1], and no angular velocity exists, the robot remains stationary without rotation.(e)Small set of feature points/Large reprojection error (B1, B4): Figure 4e demonstrates the deficiency of ORB feature points and the significant error in ORB reprojection in visual information when facing a scene with reflective glass or changing lighting. In unstructured scenes, such as glass surfaces, due to the lack of sufficient texture and detail on these surfaces, visual feature point detection algorithms have difficulty detecting a large number of feature points, which also leads to a decrease in feature points used for feature matching and camera position calculation, resulting in increased reprojection errors.(f)Large set of feature points/Small reprojection error (B2, B3): Figure 4f demonstrates the visual information has sufficient ORB feature points and a smaller ORB reprojection error when facing a no-glass scene or stable lighting scene, which is opposite to the situation illustrated in Figure 4e.

By utilizing the Wheel Odometry Fuzzy Inference System (WO-FIS) and Visual Odometry Fuzzy Inference System (VO-FIS), the system performs fuzzy calculations to predict the noise values σ^P2 and σ~P2. These predicted noise values will be applied in real time to an approximation of the process noise covariance Q¯ and observation noise covariance V¯ in (29) and (30), respectively. This approach dynamically adapts the covariance values to better reflect the current operational state, thereby enhancing the accuracy and robustness of sensor fusion in complex environments.
(29)Q¯=diag(σ^p2I3×3,σ^v2I3×3,σ^θ2I3×3,σ^ba2I3×3,σ^bα2I3×3,03)∈ℝ15×15
(30)V¯=diagσ˜p2⋅I3×3∈ℝ3×3

The fuzzification and setting fuzzy rules step is extremely crucial for FIS, and it is necessary to complete its numerical verification through pre-experiments before building the FIS system. Section 3.1 will provide a detailed description for the fuzzification and fuzzy rules determined by combining a priori experimental data. 

After that, aggregation is performed to combine these outputs into a fuzzy value [43]. In this paper, after obtaining the fuzzy output values triggered by each rule, the maximum method is used to classify all activated rules. This classification yields the combined fuzzy output values WR (*ω*) for WO-FIS and VR (*v*) for VO-FIS. The classification criterion is based on the output membership function associated with a given rule, where the maximum output fuzzy value for each membership function is calculated and then linearly combined.
(31)$$\mathrm{WR}\left(w\right)=\max\left(WR_{\left[6\right]}\left(x\right),WR_{\left[7\right]}\left(x\right)\right)+\max\left(\begin{array}{c} WR_{\left[1\right]}\left(x\right),WR_{\left[2\right]}\left(x\right),WR_{\left[3\right]}\left(x\right),\\ WR_{\left[4\right]}\left(x\right),WR_{\left[5\right]}\left(x\right),WR_{\left[8\right]}\left(x\right) \end{array}\right)$$
(32)$$VR\left(v\right)=\max\left(VR_{\left[1\right]}\left(x\right)\right)+\max\left(VR_{3}\left(x\right)\right)+\max\left(VR_{\left[2\right]}\left(x\right),VR_{\left[4\right]}\left(x\right)\right)$$where WR(x) represents the fuzzy values for any two input variables under their corresponding output membership functions (*W*_H_, *W*_L_). Similarly, VR(*x*) refers to the fuzzy values for any three input variables under their output membership functions (*V*_H_, *V*_M_, *V*_L_).

In the defuzzification process [44], the exact values of ${\hat{\sigma }}_{P}^{2}$ and ${\tilde{\sigma }}_{P}^{2}$ are obtained using the centroid method in (33).
(33)$${\hat{\sigma }}_{p}^{2}=\frac{\int w\cdot \mathrm{WR}\left(w\right)dw}{\int \mathrm{WR}\left(w\right)dw},\;{\tilde{\sigma }}_{p}^{2}=\frac{\int v\cdot \mathrm{VR}\left(v\right)dv}{\int \mathrm{VR}\left(v\right)dv}$$
where *ω*, *v* represent the fuzzy input variables for WO-FIS and VO-FIS, respectively. These two variables are used in calculating the centroid for the final defuzzied outputs.

### 2.4. The FIS-IESKF Multi-Sensor Fusion Method 

The proposed FIS-IESKF multi-sensor fusion method with the three systems is shown in Figure 5.

The odometry fusion system integrates data from a wheel encoder, an IMU and visual information from an RGBD camera. The WO supplies linear velocity data, while the IO contributes angular velocity. These data are fused using a Dead Reckoning method to compute the translational position of the robot. The rotational information is derived by integrating the angular velocity from the inertial odometry, resulting in six degrees-of-freedom (6-DoF) WIO data. For WVO, the WO provides 3-DoF position information, while the VO supplies frame-to-frame 3-DoF position as a gain term. These are linearly combined to form the 3-DoF information of the WVO.

The IESKF system is responsible for odometry estimation. The WIO and WVO data are used within the IESKF for precise odometry calculations. The WIO data are applied in the prediction phase, where the state is predicted based on the kinematic equations of the robot. The WVO data are used in the observation phase, where the state of the robot is observed according to the observation equation. During error compensation, the error between the observed and predicted states is used to adjust the state estimate, completing the odometry calculations and obtaining the updated position.

The fuzzy inference system is used to construct the WO-FIS and VO-FIS. By inputting data such as wheel speed difference, *z*-axis angular velocity, number of visual feature points, and visual reprojection error into the proposed FIS, this system dynamically predicts the process noise covariance and observation noise covariance matrices for the Iterative Error State Kalman filter. This optimization improves the accuracy of the robot’s odometry calculation, yielding more precise position and transformation.

## 3. Results of Numerical Verification and Experimental Validation

In this paper, the proposed FIS-IESKF multi-sensor fusion method is verified numerically and validated experimentally with the TWDSM robot. The experimental hardware includes an Aruco code positioning and measurement system, environmental objects, and the TWDSM robot, as shown in Figure 6. The whole TWDSM robot system consists of Kobuki base with a wheel encoder (YUJIN ROBOT Co., Ltd., Incheon, Republic of Korea), IMU (Shenzhen Yahboom Technology Co., Ltd., Shenzhen, China), RGBD camera(ASUSTeK Computer Inc., Shanghai, China), Lidar (Shanghai Slamtec Co., Ltd., Shanghai, China) and Microcomputer (NVIDIA Corporation, Santa Clara, CA, USA). Meanwhile, the Aruco code and other environment identification are supplied by TBEA Co., Ltd. from Shenyang, China.

Figure 6a illustrates the hardware setup and experimental environment configuration. The Aruco code positioning and measurement system comprises 15 Aruco codes labeled from ID-01 to ID-15, arranged in a 1-m horizontal and vertical grid pattern [45] throughout the entire experimental space. The red circles mark the robot’s start and end points. {***W***} denotes the world coordinate frame and {***R***} represents the robot’s coordinate frame, which is fixed at the robot’s chassis center. During the experiment, the robot begins at the starting point, moves along the black dashed line and ultimately returns to the starting point. This setup enables continuous assessment of effectiveness across the movement path of the robot. The primary function of this system is to provide reference positions for the predicted positions of the robot calculated by the proposed FIS-IESKF multi-sensor fusion method, which then serve as ground truth values for position measurements. These measured positions are used to assess the accuracy of the predicted positions.

The environmental object setup includes various items, with a table shown as an example in Figure 6b, placed to represent environmental objects other than the TWDSM robot. Since the RGBD camera acts as an external sensory device on the robot, additional environmental objects aid in generating visual odometry through feature point recognition and matching. The transformation relationships between the homogeneous coordinate systems in various hardware setups is shown in Figure 6b. The robot is positioned between Aruco Code ID-05 and Aruco Code ID-06. {***W***} represents the world coordinate frame. {***B***} is the robot’s base coordinate frame, which aligns with {***R***} in Figure 6a. {***I***} denotes the inertial navigation coordinate frame. {***L***} is the lidar coordinate frame. {***C***} stands for the camera coordinate frame. {***A05***} and {***A06***} are the coordinate frames for Aruco Codes ID-05 and ID-06, respectively. {***V***} represents the coordinate frame for environmental objects, with a table as a generic reference for this coordinate system in Figure 6b. Each homogeneous transformation of a hardware component and corresponding coordinate system classification and meaning are summarized in Table 3.

In the proposed FIS-IESKF method of this paper, the predicted position derived from the multi-sensor fusion system is obtained using (34).
(34)pW1=TBW⋅pB1=TVW⋅TCV⋅TLC⋅TIL⋅TBI⋅pB1
where *^W^p* and *^B^p* are the position of the robot in the world coordinate {***W***} and base coordinate {***B***}, respectively. *^W^_B_T* is the transformation matrix of the robot base coordinate frame relative to the world coordinate frame.

Similarly, the measured position calculated by the Aruco code positioning system is obtained using (35).
(35)pW1=TBW⋅pB1=TA06W⋅TCA06⋅TLC⋅TIL⋅TBI⋅pB1
where *^W^_A06_T* is the transformation matrix of the Aruco code (ID-06) coordinate frame relative to the world coordinate. Depending on the robot’s location within the experimental space, the system will match different Aruco code IDs, which define the transformation matrix between the Aruco code coordinate frame and the world coordinate frame, and apply this to the position calculation.

Using Equations (34) and (35), the predicted position and the measured position at each moment of the robot’s operation are calculated, corresponding to coordinates {***P***} and {***M***}, respectively. This setup enables the system to obtain continuous position estimations for comparison and refinement throughout the trajectory of the robot.

### 3.1. Numerical Verification for the FIS

Fuzzy membership functions [46] play a role in defining fuzzy sets and achieving fuzzification of fuzzy input variables in fuzzy logic systems. Figure 7 and Figure 8 illustrate the fuzzy membership functions of the Wheel Odometry Fuzzy Inference System (WO-FIS) and Visual Odometry Fuzzy Inference System (VO-FIS) established in this paper.

The fuzzy membership function parameters shown in Figure 7 and Figure 8 are determined by collecting and analyzing the corresponding parameter values of various motion conditions or scenarios over a period of time; the parameter range can be determined to facilitate the setting of specific fuzzy membership function values.

As shown in Figure 7, the input variables of WO-FIS include wheel speed difference
ρ, angular velocity around the *z*-axis $\omega_{e}^{z}$, and linear velocity along the *x*-axis $v_{e}^{x}$. The output variable is ${\hat{\sigma }}_{P}^{2}$. The vertical axis of the fuzzy membership functions represents the degree of membership, indicating the relationship between a specific variable and a fuzzy set across various ranges. Typically, the membership degree ranges from 0 to 1. 

To define the logical relationship between the fuzzy values of input and output variables, it is essential to set up fuzzy rules for each activated situation [47]. The fuzzy rule in this paper is shown in Table 4. *W*_H_ and *W*_L_represent the high variance and low variance membership functions, respectively. A higher variance ${\hat{\sigma }}_{P}^{2}$ indicates a lower confidence level for the noise value in the process noise covariance matrix along the diagonal entries. The rules WR_[1]_–WR_[8]_are mainly constructed by pairing the data from ${\rho ,\;\omega }_{e}^{z},\;v_{e}^{x}$ based on the kinematic state of the robot.

Regarding the selection of functional types for WO-FIS membership function, the following rules are observed in (I–VII).

(I)For A1, which represents *ρ* (+∞), the membership function is set to 1 when *ρ* approaches positive infinity and is set to 0 in all other cases. Therefore, using the “S” type membership function is appropriate for A1.(II)For A2, which represents *ρ* (0), the membership function is set to 1 when *ρ* approaches 0 and is set to 0 in all other cases. Therefore, using the “Gauss” type membership function is appropriate for A2.(III)For A3, which represents *ρ* (−∞), the membership function is set to 1 when *ρ* approaches negative infinity and is set to 0 in all other cases. Therefore, using the “Z” type membership function is appropriate for A3.(IV)For A4, which represents $\omega_{e}^{z}$ (small), the membership function is set to 0 when $\omega_{e}^{z}$ approaches positive or negative infinity and is set to 1 in all other cases. Therefore, using the “Gbell” type membership function is appropriate for A4.(V)For A5, which represents $\omega_{e}^{z}$ (large), the membership function is set to 1 when $\omega_{e}^{z}$ approaches positive or negative infinity and is set to 0 in all other cases. Therefore, using the Anti-“Gbell” type membership function is appropriate for A5.(VI)For A6, which represents $v_{e}^{x}$ (0), the membership function is set to 1 when $v_{e}^{x}$ approaches 0 and is set to 0 in all other cases. Therefore, using the “Gauss” type membership function is appropriate for A6.(VII)For all the output fuzzy inference functions of WO-FIS, our paper uses the “Triangle” type membership function for both high and low variance, as the process noise covariance matrix in the system exhibits fluctuations and variations.

The numerical results are shown in Figure 7 and Table 4. As shown in Figure 7a, when *ρ > ρ_max_*, the membership degree is 1, representing a wheel speed difference condition of *ρ* (+∞). The result of *ρ* ∈ A1 indicates that the right wheel is slipping. In rule WR_[1]_of Table 4, A1 indicates that the wheel speed difference is in state *ρ* (+∞), meaning the robot is in a right-turn slipping state. A5 shows that the angular velocity around the *z*-axis is in state $w_{e}^{z}$ (large), indicating a sharp turn.

Conversely, when *ρ < ρ_min_*, the membership degree is 1, indicating a wheel speed difference condition of *ρ* (−∞). The result of *ρ* ∈ A3 indicates that the left wheel is slipping. In addition, when *ρ*→0, the membership degree is 1, indicating a wheel speed difference condition of *ρ* (0). The result of *ρ* ∈ A2 implies that no wheel slip is occurring.

Figure 7b shows that when $\omega_{e}^{z}$ > 0.6 or $\omega_{e}^{z}$ < −0.6 for the negative range, the membership degree is 1, indicating a slow turning condition $w_{e}^{z}$ (small). When $-0.2<\omega_{e}^{z}$ < 0.2, the membership degree is 1, representing a fast-turning condition $w_{e}^{z}$ (small).

In Figure 7c, when $v_{e}^{x}$→0, the membership degree is 1, representing a state of linear velocity along the *x*-axis at $v_{e}^{x}$ (0), which implies the robot is either stationary or performing a spinning movement.

Figure 7d shows the output membership functions of WO-FIS, where the membership degree of the WO-FIS’s *W_L_* function is 1 within the range of ${\hat{\sigma }}_{P}^{2}=1/3$, and the *W_H_* function has a membership degree of 1 within the range ${\hat{\sigma }}_{P}^{2}=2/3$.

Regarding the setting of WO-FIS fuzzy rules, the following rules are followed:

i.For the WR_[1]_ and WR_[3]_ rules setting, these situations align with the kinematic state of slipping turn with a large angular velocity, which indicates that the robot is in a certain slip turning state with low confidence, and it is reasonable to give a high variance to ${\hat{\sigma }}_{P}^{2}$ for $\bar{Q}$.ii.For the WR_[2]_, WR_[4]_ and WR_[5]_ rules setting, these situations align with the kinematic state of going forward with a large angular velocity or slipping turn with a small angular velocity, which indicates that the robot is in an uncertain motion state with low confidence, and it is worth being cautious to give a high variance to ${\hat{\sigma }}_{P}^{2}$ for $\bar{Q}$.iii.For the WR_[6]_ and WR_[7]_ rules setting, these situations align with the kinematic state of going forward with a small angular velocity or stopping with a small angular velocity, which indicates that the robot is in a certain motion state with high confidence, and it is reasonable to give a low variance to ${\hat{\sigma }}_{P}^{2}$ for $\bar{Q}$.iv.For the WR_[8]_ rules setting, this situation aligns with the kinematic state of a spinning turn with a large angular velocity, which represents that the robot is in a certain motion state with low confidence, and it is reasonable to give a high variance to ${\hat{\sigma }}_{P}^{2}$ for $\bar{Q}$.

Figure 8 and Table 5 present the fuzzy membership functions and fuzzy rules of the Visual Odometry Fuzzy Inference System (VO-FIS) established in this paper. The input membership function variables for VO-FIS include the number of ORB feature points *N_u_* and the ORB feature point reprojection error *E_r_*. The output membership function variable is ${\tilde{\sigma }}_{P}^{2}$. *V*_H_, *V*_M_, *V*_L_ correspond to the high variance, medium variance, and low variance membership functions in Figure 8, respectively. A higher variance ${\tilde{\sigma }}_{P}^{2}$ indicates a lower confidence level in the noise value for the observation noise covariance matrix’s diagonal elements. The rules VR_[1]_–VR_[4]_ are constructed by combining data from *N_u_* and *E_r_* based on the status of ORB feature points in the visual camera.

Regarding the selection of functional types for VO-FIS membership functions, the following rules are observed in (VIII–XII).

(VIII)For B1, which represents *N_u_* (less), the membership function is set to 1 when *N_u_* approaches negative infinity and is set to 0 in all other cases. Therefore, using the “Z” type membership function is appropriate for B1.(IX)For B2, which represents *N_u_* (more), the membership function is set to 1 when *N_u_* approaches positive infinity and is set to 0 in all other cases. Therefore, using the “S” type membership function is appropriate for B2.(X)For B3, which represents *E_r_* (small), the membership function is set to 1 when *E_r_* approaches negative infinity and is set to 0 in all other cases. Therefore, using the “Z” type membership function is appropriate for B3.(XI)For B4, which represents *E_r_* (large), the membership function is set to 1 when *E_r_* approaches positive or negative infinity and is set to 0 in all other cases. Therefore, using the “S” type membership function is appropriate for B4.(XII)For all the output fuzzy inference functions of VO-FIS, our paper uses the “Triangle” type membership function for high, medium and low variance, as the observed noise covariance matrix in the system exhibits fluctuations and variations.

As shown in Figure 8a, the membership value of function B1 is 1 within the range *N_u_* < $N_{u}^{\min}$, representing a condition where the number of ORB feature points is low *N_u_*(less). In contrast, the membership value of function B2 is 1 within the range *N_u_* > $N_{u}^{\max}$, indicating a state where the ORB feature points are abundant *N_u_* (more).

Figure 8b demonstrates that the membership value of function B3 is 1 when the ORB reprojection error *E_r_* < $E_{r}^{\min}$, reflecting a low reprojection error for ORB feature points *E_r_* (small). On the other hand, function B4 has a membership value of 1 within the range *E_r_* > $E_{r}^{\max}$, indicating a high ORB feature point reprojection error *E_r_* (large).

In Figure 8c, the output membership functions of VO-FIS are shown. The membership function *V*_L_ takes a value of 1 when ${\tilde{\sigma }}_{P}^{2}=1/4$, *V*_M_ takes a value of 1 when ${\tilde{\sigma }}_{P}^{2}=1/2$, and *V*_H_ takes a value of 1 when ${\tilde{\sigma }}_{P}^{2}=3/4$.

Regarding the setting of VO-FIS fuzzy rules, the following rules are followed:i.For the VR_[1]_ rules setting, this situation aligns with the visual information of a low reprojection error with a large number of key points, which indicates that the robot is in a high quality environment scene with high confidence, and it is reasonable to give a low variance to ${\tilde{\sigma }}_{P}^{2}$ for $\bar{V}$.ii.For the VR_[3]_ rules setting, this situation aligns with the visual information of a low reprojection error with a small number of key points, which indicates that the robot is in a medium quality environment scene with medium confidence, and it is worth being cautious to give a medium variance to ${\tilde{\sigma }}_{P}^{2}$ for $\bar{V}$.iii.For the VR_[2]_ and VR_[4]_ rules setting, these situations align with the visual information of a large reprojection error, which indicates that the robot is in a low quality environment scene with low confidence, and it is worth being cautious to give a high variance to ${\tilde{\sigma }}_{P}^{2}$ for $\bar{V}$.

Let WR_[λ]_[*x*] (λ = 1, 2 …8) and WR_[τ]_[*x*] (τ = 1, 2 …4) represent the output fuzzy values for individual rules in WR_[1]_–WR_[8]_ and VR_[1]_–VR_[4]_, respectively. The aggregated output fuzzy value can be obtained for each individual rule in Table 6.

Therefore, the specific inputs in the proposed FIS system can be assigned corresponding fuzzy values based on the membership functions established, thus enabling the fuzzification process.

### 3.2. Experimental Validation for the FIS-IESKF Multi-Sensor Fusion Method

In this paper, the proposed FIS-IESKF method is compared with the EKF and the ground truth trajectory values captured by visual sensors detecting each Aruco marker shown in Figure 6. Since the proposed FIS-IESKF method incorporates three types of sensors (wheel encoders, an IMU and a visual sensor), the EKF method is also configured to fuse the odometry information from these three sensors, ensuring consistency in hardware and algorithmic inputs. Furthermore, the experimental scenarios include straight motion, sharp turns, slow turns, rotational movement, movement in a scene with glass, movement in a non-glass scene, movement in a changing lighting scene and movement in a stable lighting scene. Each scenario corresponds to the relevant fuzzy values shown in Table 7. The experimental results consist of two main parts: (1) trajectory similarity comparison between FIS-IESKF, EKF, UKF and the ground truth; (2) trajectory error assessment. The robot in the experiment is set to move in a closed-loop motion from start to end points. The global and local error between the evaluated trajectory and the ground truth are compared. Absolute Position Error (APE) and Relative Position Error (RPE) are used to analyze the evaluated trajectories for FIS-IESKF.

The criterion of Fréchet Distance [48] is used to assess the trajectory similarity in this paper by (36)–(38).
(36)$$D_{F}\left(A,B\right)=\underset{\alpha ,\beta }{\inf}\underset{t\in \left[0,1\right]}{\max}|A\left(\alpha \left(t\right)\right)-B\left(\beta \left(t\right)\right)|$$
(37)$$D_{F}\left(A,C\right)=\underset{\alpha ,\gamma }{\inf}\underset{t\in \left[0,1\right]}{\max}|A\left(\alpha \left(t\right)\right)-C\left(\gamma \left(t\right)\right)|$$
(38)$$D_{F}\left(A,D\right)=\underset{\alpha ,\psi }{\inf}\underset{t\in \left[0,1\right]}{\max}|A\left(\alpha \left(t\right)\right)-D\left(\psi \left(t\right)\right)|$$where *α*(*t*), *β*(*t*), *γ*(*t*), *ψ*(*t*) are the reparameterization functions for trajectories A, B, C, D, respectively. $\max_{t\in [0,\;1]}$ denotes the maximum point-to-point distance between the two curves at all times *t* under a certain parameterization condition. The objective is to find a pair of parameterization functions (inf*_α,β_*, inf*_α,γ_*, inf*_α,ψ_*) that minimizes the maximum distance across all possible pairs of parameterization functions in Figure 9. As illustrated in Figure 9, trajectory A represents the ground truth, while trajectory B represents the estimated trajectory from the FIS-IESKF, trajectory C represents the estimated trajectory from the EKF, and trajectory D represents the estimated trajectory from the UKF.

The APE and RPE in results comparison is defined as (39) and (40).
(39)$${\mathrm{APE}}_{i}=\|\log\left(T_{\mathrm{ref},i}^{-1}T_{\mathrm{esti},i}\right)\|$$
(40)$${\mathrm{RPE}}_{i}=\|\log\left({\left(T_{gt,i}^{-1}T_{gt,i+\Delta t_{f}}\right)}^{-1}\left(T_{\mathrm{esti},i}^{-1}T_{\mathrm{esti},i+\Delta t_{f}}\right)\right)\|$$
where APE is used to compute the absolute position error for each frame [49], while RPE calculates the error between adjacent frames [50].

Figure 10 presents trajectory similarity comparison results for different scenarios: straight motion, spin turning, slow turning and sharp turning. As shown in Figure 10, the trajectory estimated by the FIS-IESKF algorithm closely follows the ground truth trajectory. In contrast, the trajectory estimated by the EKF algorithm gradually exhibits yaw drift over time during low-speed straight and slow turning motions. During spin turning and sharp turning movements, significant yaw drift occurs primarily at the turning points, leading to considerable global odometry displacement. The trajectory estimated by the UKF exhibited less yaw drift over time during low-speed straight and slow turning motions compared to the EKF. Additionally, during spin turning and sharp turning maneuvers, the trajectory deviation of the UKF was significantly reduced compared to the EKF, although it remained inferior to that of the FIS-IESKF.

Figure 11 and Figure 12 illustrate the trajectory comparison and actual scene images for scenarios with and without glass obstacles, which will prove the adaptability of the FIS-IESKF algorithm to unstructured environments. In Figure 11, when encountering a scene with glass, the number of paired feature points remains between 50 and 70. In this case, it exhibits substantial errors and minor error when the EKF and UKF algorithms estimate the robot’s turning position, respectively, while the FIS-IESKF algorithm maintains a position that is roughly consistent with the ground truth.

In Figure 12, when no glass obstacles are present, the number of paired feature points ranges from 200 to 270. Here, the position estimated by the EKF method has significant fluctuations, while the position estimated by the FIS-IESKF algorithm remains approximately aligned with the ground truth. The position estimated by the UKF algorithm does not exhibit significant errors, although it still performs worse than the FIS-IESKF method.

Figure 13 presents the trajectory comparisons and actual scene images for scenarios involving both changing and stable lighting conditions, demonstrating the adaptability of the proposed FIS-IESKF to varying lighting environments. To minimize the influence of non-illumination factors, the experiments were conducted in a low-speed motion mode. As illustrated in Figure 13a, significant changes in the light source cause a sharp decline in the number of ORB feature points, leading to notable trajectory fluctuations in the estimations provided by the EKF and UKF. In contrast, the trajectory estimated by the proposed FIS-IESKF method remains as consistent as possible with the ground truth. Under stable lighting conditions, shown in Figure 13b, none of the algorithms experience a sharp drop in ORB feature points or significant trajectory fluctuations, indicating stable performance. Therefore, the proposed FIS-IESKF is proved to have the ability to ensure accurate positioning even under environmental variability.

The results presented in Figure 9, Figure 10, Figure 11, Figure 12 and Figure 13 are summarized in Table 8. The evaluation of the Fréchet Distance values, denoted as *D_F_* (A, B), *D_F_* (A, C) and *D_F_* (A, D), across various experimental categories enables a comparative analysis of the trajectory estimations produced by the FIS-IESKF, EKF and UKF against the ground truth trajectory.

It can be found from Table 8 that *D_F_* (A, B) < *D_F_* (A, D) < *D_F_* (A, C) holds true across all six motion categories, indicating that the trajectories estimated using the proposed FIS-IESKF method exhibit a higher similarity to the ground truth. The ratio *D_F_* (A, C)/*D_F_* (A, B) reveals that as the robot’s speed increases, along with the occurrence of turning maneuvers, glass scenes and the changing lighting scene, the values of *D_F_* (A, C)/*D_F_* (A, B) rise significantly. Furthermore, the ratio *D_F_* (A, D)/*D_F_* (A, B) indicates that the UKF maintains trajectory-following stability even in sharp turning motion, spin motion, the glass scene and the changing lighting scene. The average value mean (*D_F_* (A, C)/*D_F_* (A, B)) and mean (*D_F_* (A, D)/*D_F_* (A, B)) reaches 5.316 and 2.485, respectively. This indicates that, in terms of trajectory similarity assessment, the FIS-IESKF method enhances tracking performance relative to the ground truth by approximately 531% and 248% compared to the EKF and UKF, respectively.

The experimental results of the global trajectory position error and local adjacent frame position error is shown in Figure 14. The TWDSM robot is set to conduct a closed-loop long-distance motion from start to end. The experimental trajectories are compared by the APE and RPE.

Figure 14 illustrates the results of the closed-loop motion assessment of the robot. In the APE evaluation, the FIS-IESKF algorithm demonstrated good adherence to the ground truth, exhibiting a smaller absolute position error. Conversely, the EKF algorithm showed considerable fluctuations in absolute position error compared to the ground truth, particularly evident during turning maneuvers. Meanwhile, the trajectory estimated by UKF exhibited more severe global position errors, especially during turning maneuvers. In the RPE evaluation, the FIS-IESKF algorithm achieved relatively low errors compared to the ground truth, confirming its superior inter-frame odometry performance. However, the EKF and UKF algorithm did not display more stable relative errors with respect to the ground truth.

The root mean square error (RMSE) serves as a crucial metric for evaluating the robot’s motion position using the APE and RPE, reflecting the overall magnitude of the errors in Table 9 and Table 10. The evaluation results of APE and RPE encompass assessments of both translational and rotational motion.

The results indicate that the translational estimation performance of the FIS-IESKF, EKF, and UKF algorithms stays within a reasonable range. Specifically, the FIS-IESKF shows the best translational accuracy with an RMSE of 0.009396 m, followed by the UKF at 0.016228 m, and the EKF at 0.054120 m, which is notably higher. In terms of rotational estimation, the EKF and UKF display significant deviations from the ground truth, with RMSEs of 64.3963° and 81.3227°, respectively. On the other hand, the FIS-IESKF achieves superior overall accuracy, with a global rotational RMSE of 0.149480°. Two probable reasons contribute to this problem: (1) Nonlinear effects and error accumulation can cause the EKF and UKF to gradually deviate from the actual values during long-duration rotations or complex paths, leading to less accurate position estimation. (2) These algorithms may struggle to accurately predict the covariance in highly nonlinear systems, which limits their estimation performance.

Additionally, the FIS-IESKF demonstrates impressive local adjacent frame performance, achieving a translational RMSE of 0.002347 m and a rotational RMSE of 0.201990°. These results significantly outperform those of both the EKF and UKF algorithms, confirming the FIS-IESKF’s robustness in maintaining high trajectory accuracy under long-distance closed-loop motion.

### 3.3. Computational Complexity Analysis for the FIS-IESKF Multi-Sensor Fusion Method

The computational complexity analysis method is an essential way to evaluate the real-time performance of SLAM algorithms. The steps of the FIS-IESKF algorithm largely follow the overall framework of the IESKF, with the key difference being that during the IESKF prediction step, it incorporates covariance prediction using WO-FIS, and during the observation step, it integrates covariance prediction using VO-FIS.

In terms of theoretical calculation for the IESKF method, our paper decomposes the complexity of each key step and combines the properties of matrix operations to obtain the overall asymptotic computational complexity as ①–⑤. It should be noted in advance that *N* represents the dimension of the state vector while *M* represents the dimension of the observation vector. Besides, due to the nominal status **x** and observation **z** having dimensions of 18 × 1 and 3 × 1, the value of *N* and *M* is defined as 18 and 3, respectively.

① In the IESKF initialization step, the primary contribution to the asymptotic computational complexity comes from the covariance matrix initialization ($\hat{P_{t}}$.) in Equation (13). This matrix consists of six diagonal blocks, each represented as a 3 × 3 matrix, which involves asymptotic computational complexity of the IESKF initialization step which is 6 × *O*(3 × 3) = *O*(54).

② In the IESKF prediction step, the primary contribution to the asymptotic computational complexity comes from the prior estimate covariance calculation in Equation (17). **F_x_** and **P***_t_* both have a dimension of *N* × *N*, therefore the asymptotic computational complexity of matrices **F_x_** and **P***_t_* is *O*(*N*^2^) and matrix multiplication is performed in this formula. The overall asymptotic computational complexity is *O*(*N*^3^) = *O*(5832).

③ In the IESKF observation step, the Kalman gain (**K**) computation is the primary contributor to the overall asymptotic computational complexity in Equation (25). **H**, **V** and $P_{t}^{-}$ have dimensions of *M* × *N*, *M* × *M* and *N* × *N*, respectively, which involves calculating **H**$P_{t}^{-}$ through matrix multiplication with a complexity of *O*(*MN*^2^), and performing a matrix inversion for **H**$P_{t}^{-}$**H***^T^* + **V**, which has a complexity of *O*(*M*^3^). Combining these operations, the overall asymptotic computational complexity of the observation step is *O*(*MN*^2^ + *M*^3^) = *O*(999).

④ In the IESKF error compensation step, the computation of error state compensation is the primary contributor to the overall asymptotic computational complexity. Specifically, when performing element-wise addition as described in Equation (28), the complexity is *O*(*N*) = *O*(18).

⑤ In the IESKF error reset step, the asymptotic computational complexity of setting the error state to zero (δ**x** = **0**_18×1_) is *O*(*N*) = *O*(18), as each of the *N* elements must be individually assigned to zero.

In terms of theoretical calculation for WO-FIS and VO-FIS, the calculation of asymptotic computational complexity follows Equation (41) as shown below [51]. Therefore, the asymptotic computational complexity of WO-FIS and VO-FIS are shown in ⑥ and ⑦, respectively.
(41)$$O_{\mathrm{FIS}}=O\left(\sum_{j=1}^{N_{input}}F_{j}\right)+O\left(R\cdot \left(N_{input}+N_{output}\right)\right)+O\left(R\cdot N_{output}\right)$$
where *N_input_* and *N_output_* represent the number of input/output variables, *F*_j_ represents the number of membership functions for a single input, and *R* represents the number of rules in the FIS system.

⑥ In the WO-FIS, owing to *N_input_* = 3, *N_outout_* = 1 and *R* = 8, meanwhile *F*_1_ = 3, *F*_2_ = 2 and F_3_ = 1 (representing the number of membership functions for *ρ*, $\omega_{e}^{z}$, $v_{e}^{x}\;$, respectively), the asymptotic computational complexity of WO-FIS is *O*_WO-FIS_ = *O*(*F*_1_ + *F*_2_ + *F*_3_) + *O*(*R*·(*N_input_* + *N_outout_*)) + *O*(*R*·*N_outout_*) =*O*(6) + *O*(32) + *O*(8) = *O*(46).

⑦ In the WO-FIS, owing to *N_input_* = 2, *N_outout_* = 1 and *R* = 4, meanwhile *F*_1_ = 2 and *F*_2_ = 2 (representing the number of membership functions for *N_u_* and *E_r_*, respectively), the asymptotic computational complexity of VO-FIS is *O*_VO-FIS_ = *O* (*F*_1_ + *F*_2_) + *O* (*R*·(*N_input_* + *N_outout_*)) + *O*(*R*·*N_outout_*) = *O*(4) + *O*(12) + *O*(4) = *O*(20).

As a result, the computational complexity of the FIS-IESKF method has been summarized in Table 11. It is noted that *Ts* represents the time step. Due to the fact that, in FIS-IESKF, *Ts* depends on the sampling frequency and system operation time after sensor time synchronization is applied, when the sampling frequency is set to 20 Hz and the system operation time is 100 s, Ts takes a value of 2000. Therefore, the time proportion of each step in Table 11 was calculated using Equation (42).
(42)$$C_{tp}=\left(C_{k}/C_{total}\right)\cdot 100\%$$
where *C_k_* represents each specific computational complexity value in Table 11, and *C_total_* represents the sum of each *C_k_*.

The computational complexity analysis shows that the IESKF prediction step dominates the total time with 84.11%, followed by the IESKF observation step at 14.41%, as both involve computationally intensive matrix operations (*O*(*N*^3^) and *O*(*MN*^2^ + *M*^3^), respectively). The error compensation, error reset and other minor steps contribute minimally (<1%) due to simpler operations like O(*N*^2^). Notably, the inclusion of the FIS system has a negligible impact on the overall runtime of the FIS-IESKF system, as FIS-specific steps such as prediction with FIS and observation with FIS contribute less than 1% of the total time. This demonstrates that FIS integration does not significantly slow down the overall performance of the algorithm.

## 4. Conclusions

A multi-sensor fusion method using a fuzzy inference system (FIS) in the wheel-inertial-visual odometry (WIVO) system for position estimation of two-wheel differential speed mobile (TWDSM) robots in indoor environments is established in this paper. Due to the dynamic characteristics of the system, applying fixed covariance matrices for process noise and observation noise in the iterative Kalman filter algorithm is insufficient to adapt effectively to the complex kinematic characteristics faced by the robot, as well as the intricate impacts of changing lighting conditions and unstructured scenes in the camera model on visual observation models. This paper provides a metrology for inferring the fusion Kalman filter gain using a fuzzy inference system, establishing fuzzy inference systems for both wheeled odometry and visual odometry across various scenarios. This approach allows for dynamic predictions of the system covariance matrix noise.

Through experiments across eight distinct scenarios (straight motion, sharp turning, slow turning, spin motion, environments with and without glass, and changing or stable lighting), we demonstrate the effectiveness of the FIS-IESKF method by comparing it to the EKF, UKF and ground truth trajectories obtained through Aruco markers. In terms of trajectory similarity assessment, the tracking performance of the FIS-IESKF algorithm is on average approximately 542% better than that of the EKF algorithm relative to the ground truth. This indicates that the FIS-IESKF method produces trajectories that more closely match the ground truth across various motion types, including straight, turning and spin motions, as well as in complex environments like glass and changing lighting scenes.

Additionally, results from closed-loop movement tests of the robot indicate that the FIS-IESKF algorithm significantly enhances the performance of rotational estimation, with a global rotational RMSE of 0.149°, far outperforming the EKF (64.3963°) and UKF (81.3227°). The global translational RMSE of the FIS-IESKF is less than 0.01 m, while the EKF and UKF show higher errors of 0.0541 m and 0.0162 m, respectively. The FIS-IESKF also achieves superior local adjacent frame performance with a translational RMSE of 0.0023 m and rotational RMSE of 0.2020°, outperforming both EKF and UKF in maintaining trajectory accuracy in long-distance closed-loop motion.

In terms of adaptability to complex environments, the FIS-IESKF method shows higher robustness in varying environments. In glass scenes, FIS-IESKF’s position estimation remains closely aligned with the ground truth, while both EKF and UKF exhibit significant deviations. Similarly, in scenarios with changing lighting conditions, the FIS-IESKF demonstrates higher consistency, while the EKF and UKF exhibit considerable fluctuations in trajectory.

In terms of analysis to computational complexity, the inclusion of the FIS system has negligible impact on the runtime of the FIS-IESKF algorithm, with FIS covariance prediction steps contributing less than 1% of the total time. Therefore, the proposed method is validated to show improved robustness and accuracy of multi-sensor fusion localization for two-wheeled differential drive indoor mobile robots.

## Figures and Tables

**Figure 1 sensors-24-07619-f001:**
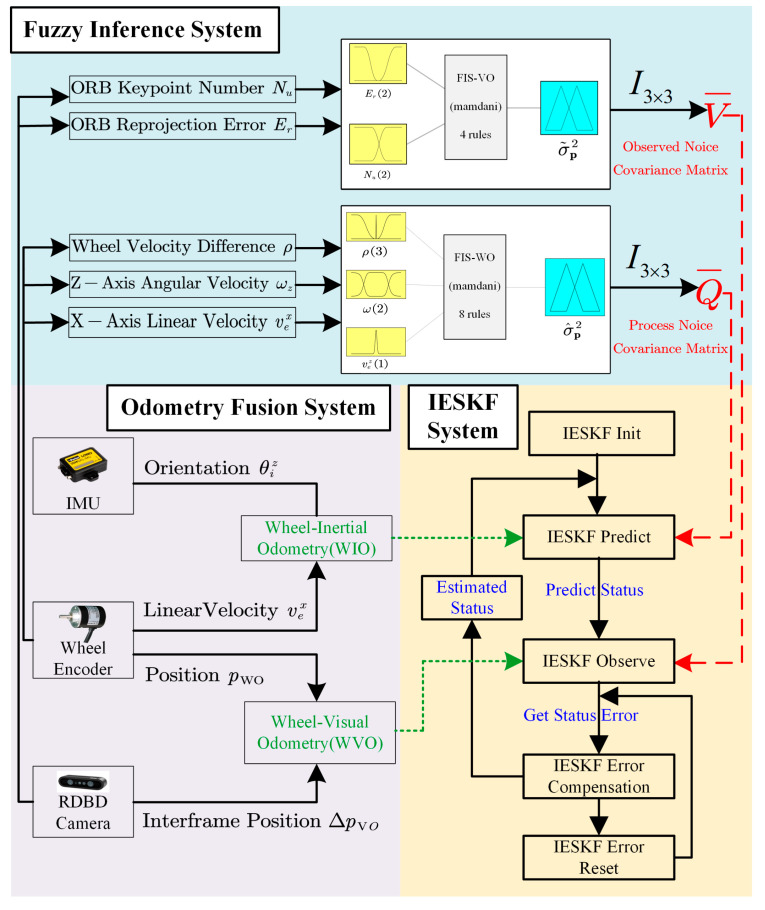
Multi-sensor fusion system.

**Figure 2 sensors-24-07619-f002:**
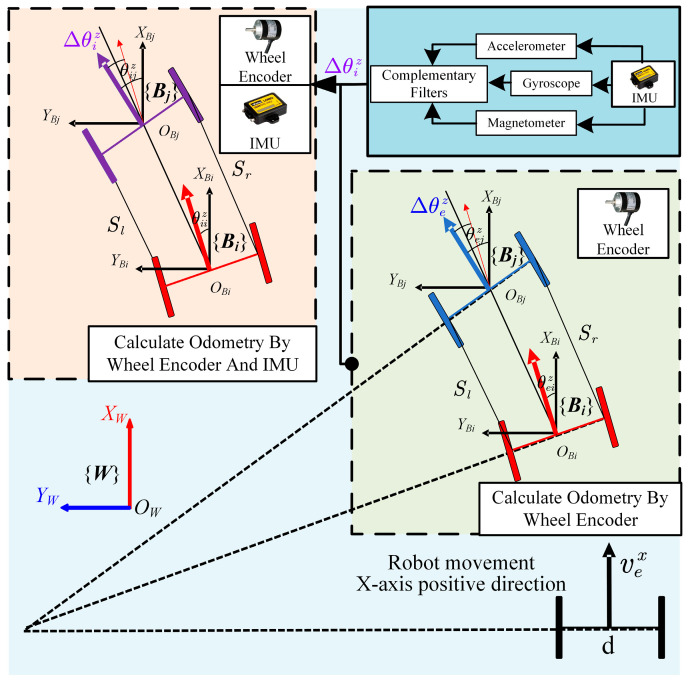
Formulation of the Wheel-Inertial Odometry in TWDSM robot.

**Figure 3 sensors-24-07619-f003:**
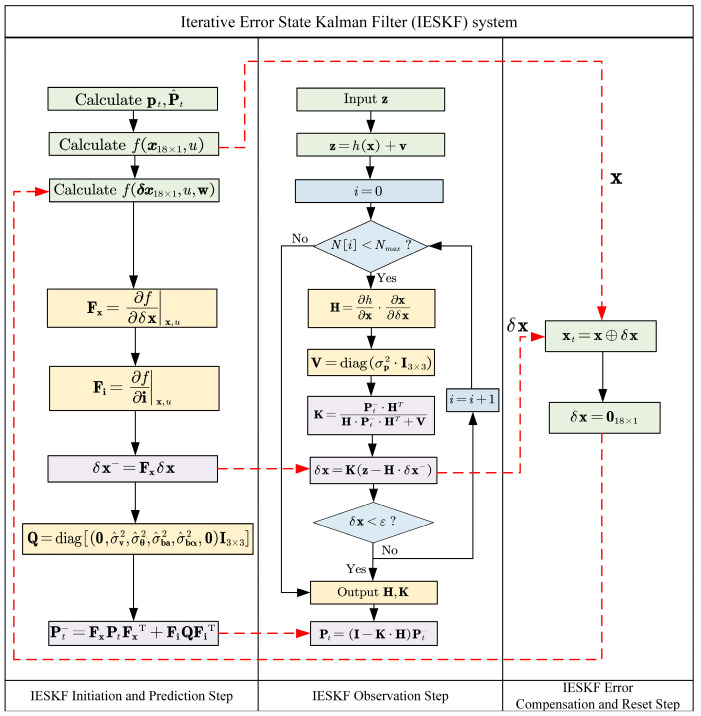
Flowchart of the IESKF system.

**Figure 4 sensors-24-07619-f004:**
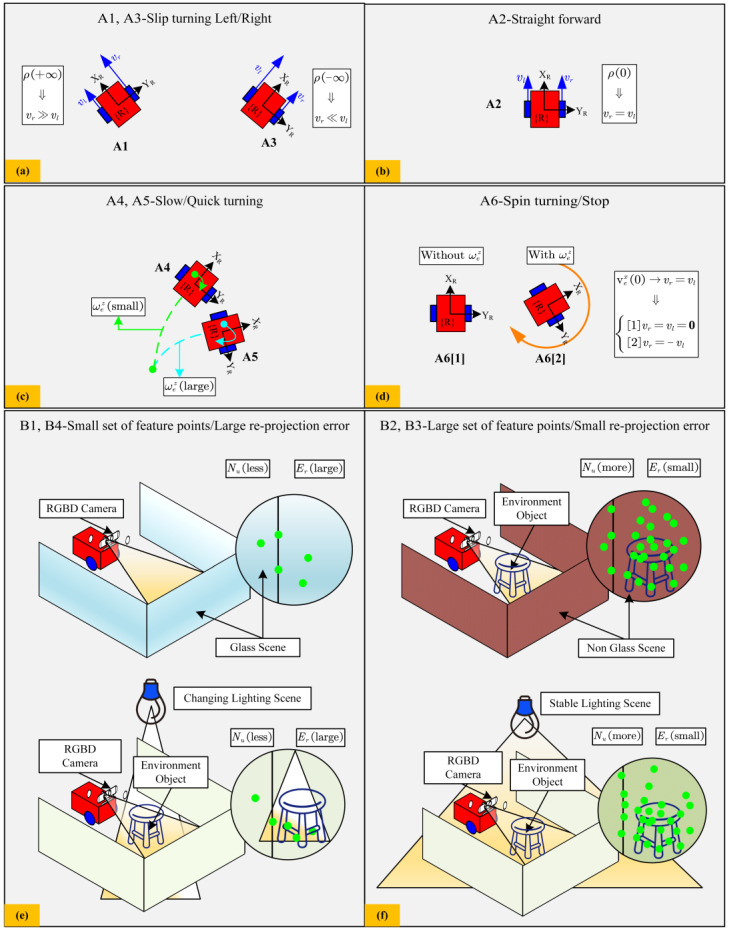
The physical meanings of the fuzzy inference system utilizing wheel odometry and visual odometry information.(**a**) Slip turning in left or right, (**b**) Straight forward, (**c**) Turn with slow or quick angular acceleration, (**d**) Spin turning or stop, (**e**) Unstructured scene and varying light scene and (**f**) Structured scene and stable lighting scene.

**Figure 5 sensors-24-07619-f005:**
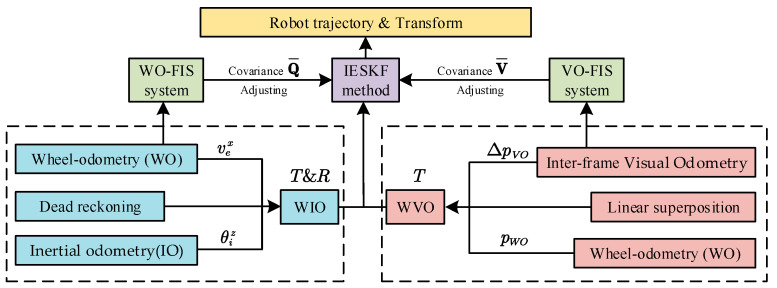
The diagram of FIS-IESKF multi-sensor fusion method.

**Figure 6 sensors-24-07619-f006:**
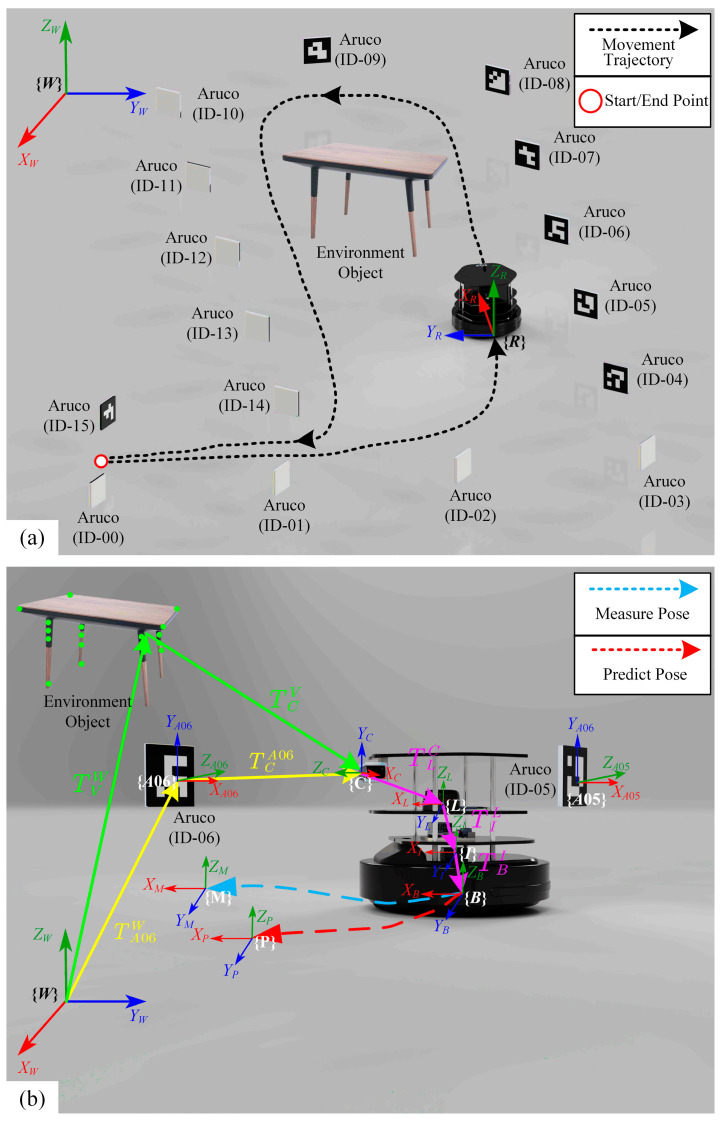
System hardware for the experiment. (**a**) Experimental environment settings, and (**b**) transformation relationships of homogeneous coordinate systems in various systems.

**Figure 7 sensors-24-07619-f007:**
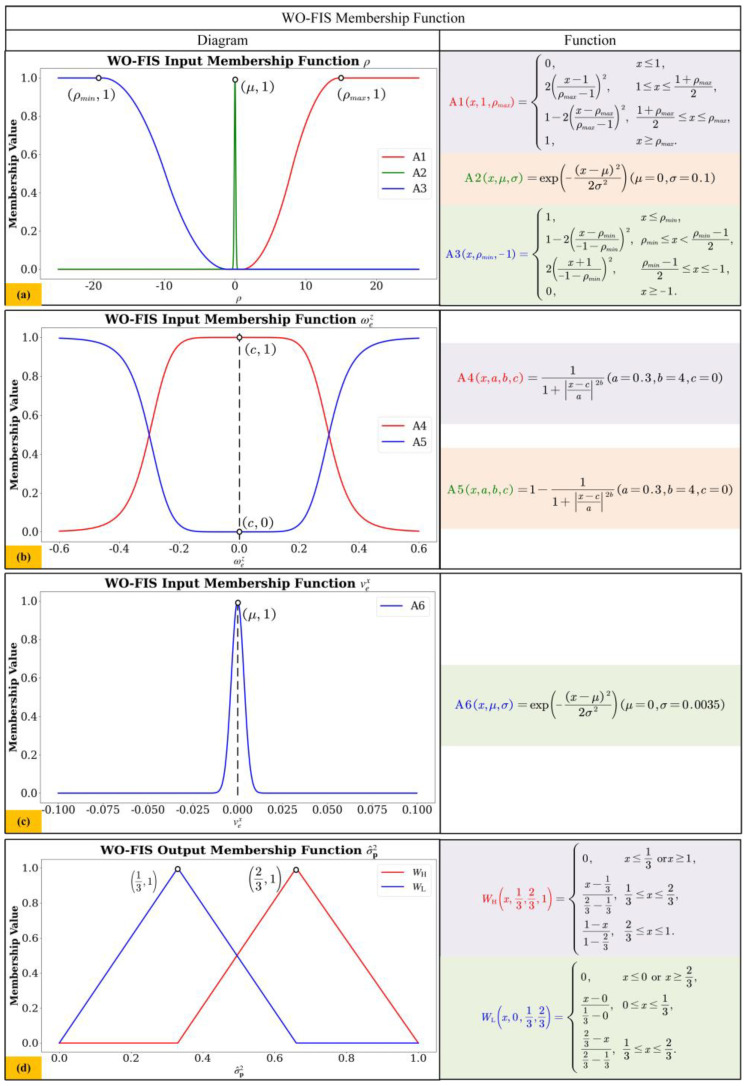
WO-FIS membership functions. (**a**) Input WO-FIS membership function for *ρ*, (**b**) Input WO-FIS membership function for $\omega_{e}^{z}$, (**c**) Input WO-FIS membership function for $v_{e}^{x}$ and (**d**) Output WO-FIS membership function.

**Figure 8 sensors-24-07619-f008:**
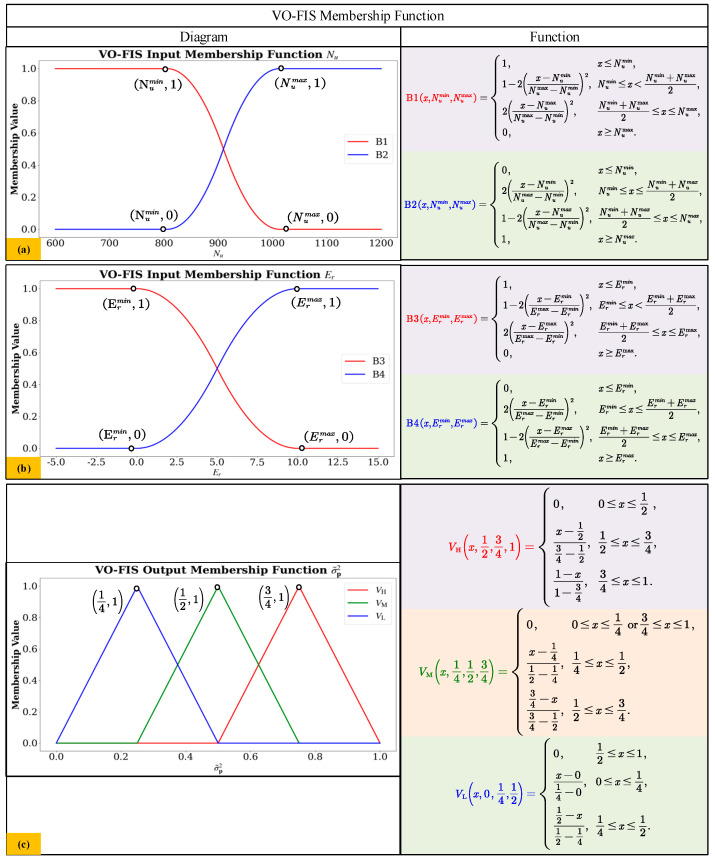
VO-FIS membership functions. (**a**) Input VO-FIS membership function for *N_u_*, (**b**) Input VO-FIS membership function for *E_r_* and (**c**) Output VO-FIS membership function.

**Figure 9 sensors-24-07619-f009:**
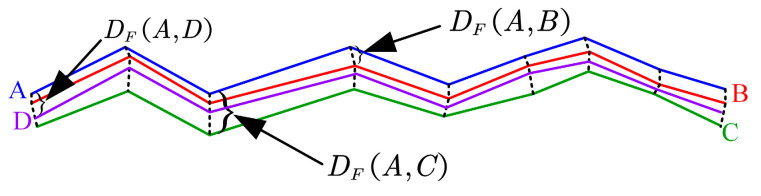
The maximum value of the paths generated by the FIS-IESKF, EKF and UKF algorithms under optimal parameterization compared to the ground truth.

**Figure 10 sensors-24-07619-f010:**
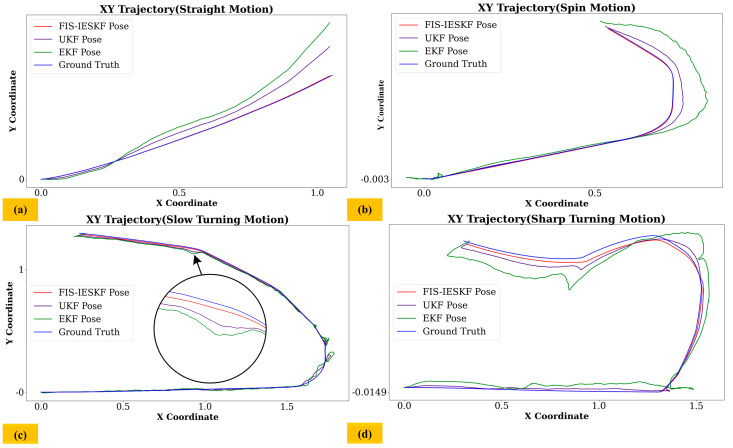
Trajectory comparison diagram of straight motion, sharp turning, slow turning and spin-turning. (**a**) Straight motion, (**b**) Spin motion, (**c**) Slow turning motion and (**d**) Sharp turning motion.

**Figure 11 sensors-24-07619-f011:**
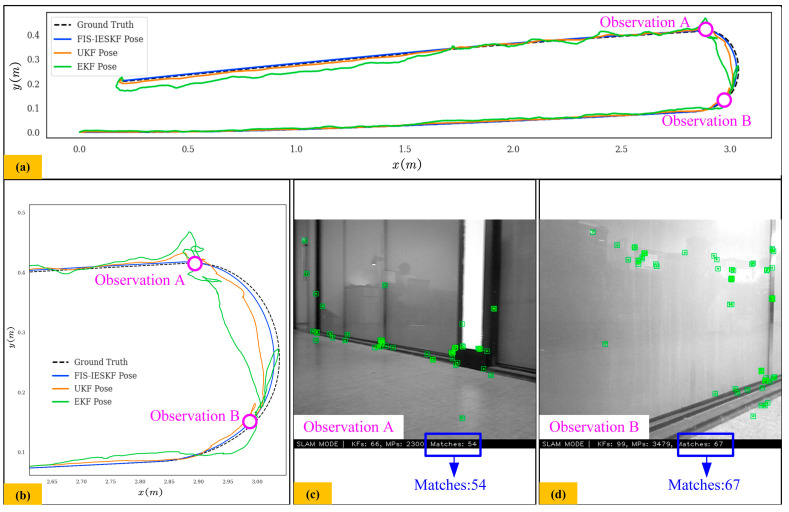
Trajectory comparison in a scene with glass. (**a**) Global motion trajectory, (**b**) Local motion trajectories at bends, (**c**) Visual feature point situation at Observation A and (**d**) Visual feature point situation at Observation B.

**Figure 12 sensors-24-07619-f012:**
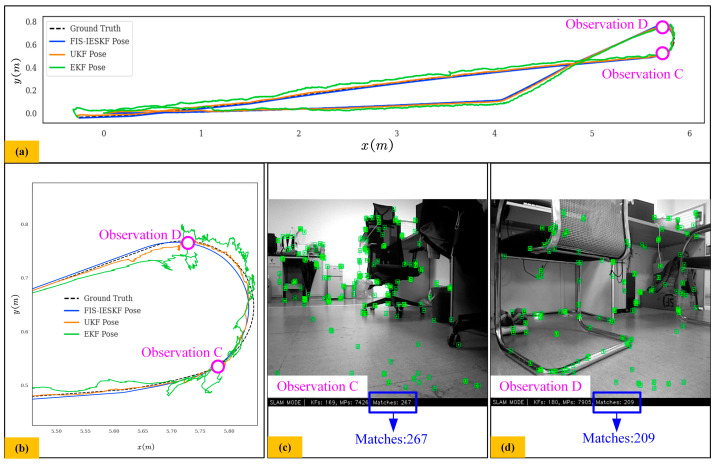
Trajectory comparison in non-glass scene. (**a**) Global motion trajectory, (**b**) Local motion trajectories at bends, (**c**) Visual feature point situation at Observation C and (**d**) Visual feature point situation at Observation D.

**Figure 13 sensors-24-07619-f013:**
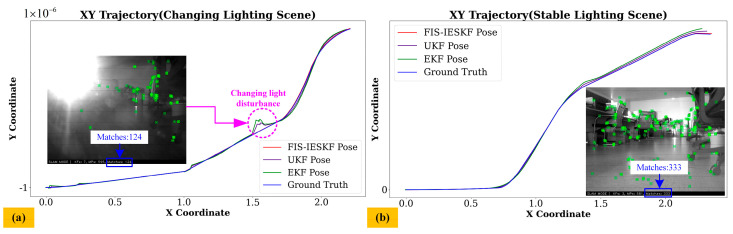
Trajectory comparison in changing lighting scene and stable lighting scene. (**a**) Changing lighting scene and (**b**) Stable lighting scene.

**Figure 14 sensors-24-07619-f014:**
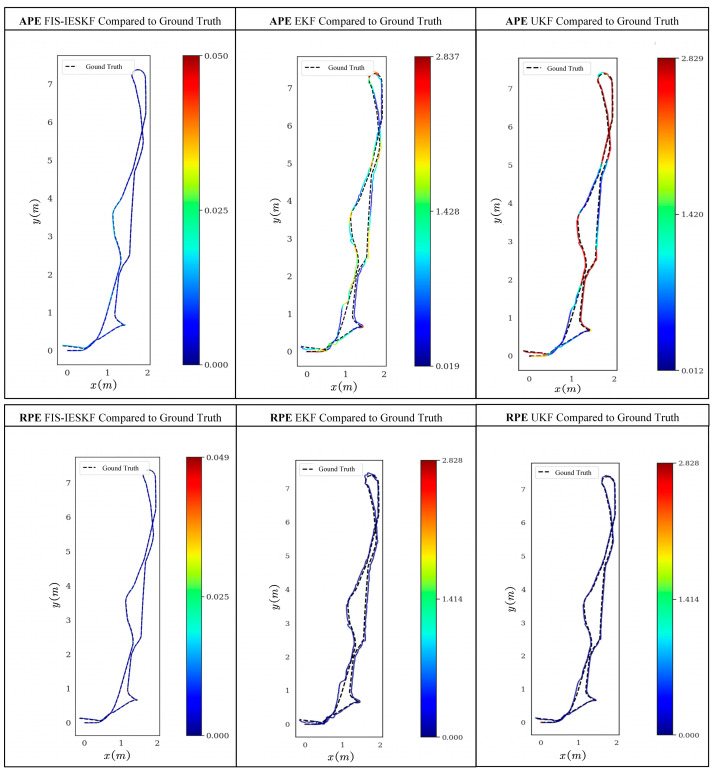
The result of closed-loop motion evaluated by APE and RPE.

**Table 1 sensors-24-07619-t001:** Status variable and status type in IESKF method.

Status Variable	Status Types		
	True Status (x*_t_*)	Nominal Status (x)	Error Status (δx)
Position	**p** * _t_ *	**p**	δ**p**
Velocity	**v** * _t_ *	**v**	δ**v**
Orientation	**θ** * _t_ *	**θ**	δ**θ**
Linear acceleration bias	**b_a_** * _t_ *	**b_a_**	δ**b_a_**
Angular acceleration bias	**b_α_** * _t_ *	**b_α_**	δ**b_α_**
Gravitational acceleration	**g** * _t_ *	**g**	δ**g**

**Table 2 sensors-24-07619-t002:** The proposed fuzzy set and the running effects on the robot.

Fuzzy Set	Variable	Running Effects
A1	*ρ* (+∞)	Slip turning Left
A2	*ρ* (0)	Straight forward
A3	*ρ* (−∞)	Slip turning Right
A4	$\omega_{e}^{z}$ (small)	Slow turning
A5	$\omega_{e}^{z}$ (large)	Quick turning
A6	$v_{e}^{x}$ (0)	Spin turning/Stop
B1	*N_u_* (less)	Small set of feature points
B2	*N_u_* (more)	Large set of feature points
B3	*E_r_* (small)	Small reprojection error
B4	*E_r_* (large)	Large reprojection error

**Table 3 sensors-24-07619-t003:** Homogeneous transformation coordinate system.

Homogeneous Coordinate System	Meaning
TVW	The environment object coordinate system is relative to the world coordinate system
TCV	The camera coordinate system is relative to the environment object coordinate system
TA06W	The Aruco code (ID-06) coordinate system is relative to the world coordinate system
TCA06	The camera coordinate system is relative to the Aruco code (ID-06) coordinate system
TLC	The lidar coordinate system is relative to the camera coordinate system
TIL	The inertial navigation coordinate system is relative to the lidar coordinate system

**Table 4 sensors-24-07619-t004:** The fuzzy rules of WO-FIS.

Rules	*ρ*	$\omega_{e}^{z}$	$v_{e}^{x}$	${\hat{\sigma }}_{P}^{2}$
WR_[1]_	A1	A5	**/**	*W* _H_
WR_[2]_	A2	A5	**/**	*W* _H_
WR_[3]_	A3	A5	**/**	*W* _H_
WR_[4]_	A1	A4	**/**	*W* _H_
WR_[5]_	A3	A4	**/**	*W* _H_
WR_[6]_	A2	A4	**/**	*W* _L_
WR_[7]_	**/**	A4	A6	*W* _L_
WR_[8]_	**/**	A5	A6	*W* _H_

**Table 5 sensors-24-07619-t005:** The fuzzy rules of VO-FIS.

Rules	*N_u_*	*E_r_*	${\tilde{\sigma }}_{P}^{2}$
VR_[1]_	B2	B3	*V* _L_
VR_[2]_	B2	B4	*V* _H_
VR_[3]_	B1	B3	*V* _M_
VR_[4]_	B1	B4	*V* _H_

**Table 6 sensors-24-07619-t006:** The fuzzy output under single rule.

Rules	Fuzzy Output
WR_[1]_	WR_[1]_(*x*) = *min*[A1(*x*), A5(*x*)]
WR_[2]_	WR_[2]_(*x*) = *min*[A2(*x*), A5(*x*)]
WR_[3]_	WR_[3]_(*x*) = *min*[A3(*x*), A5(*x*)]
WR_[4]_	WR_[4]_(*x*) = *min*[A1(*x*), A4(*x*)]
WR_[5]_	WR_[5]_(*x*) = *min*[A3(*x*), A4(*x*)]
WR_[6]_	WR_[6]_(*x*) = *min*[A2(*x*), A4(*x*)]
WR_[7]_	WR_[7]_(*x*) = *min*[A4(*x*), A6(*x*)]
WR_[8]_	WR_[8]_(*x*) = *min*[A5(*x*), A6(*x*)]
VR_[1]_	VR_[1]_(*x*) = *min*[B2(*x*), B3(*x*)]
VR_[2]_	VR_[2]_(*x*) = *min*[B2(*x*), B4(*x*)]
VR_[3]_	VR_[3]_(*x*) = *min*[B1(*x*), B3(*x*)]
VR_[4]_	VR_[4]_(*x*) = *min*[B1(*x*), B4(*x*)]

**Table 7 sensors-24-07619-t007:** Experimental comparison.

Experiment and the Fuzzy Set	Compared Methods			
	FIS-IESKF (Our Method)	EKF	UKF	Ground Truth
Straight Motion	A2			
Sharp Turning Motion	A1/A3/A5			
Slow Turning Motion	A4			
Spin Motion	A6			
Glass Scene	B1/B4			
Changing Lighting Scene				
Non-Glass Scene	B2/B3			
Stable Lighting Scene				

**Table 8 sensors-24-07619-t008:** Trajectory similarity comparison.

Experiment	*D_F_* (A, B)	*D_F_* (A, C)	*D_F_* (A, D)	*D_F_* (A, C)/*D_F_* (A, B)	*D_F_* (A, D)/*D_F_* (A, B)
Straight Motion	0.009156	0.013778	0.010814	1.504756288	1.181044745
Sharp Turning Motion	0.021493	0.103382	0.039939	4.809883826	1.858172395
Slow Turning Motion	0.015056	0.057390	0.034751	3.811694695	2.308055046
Spin Motion	0.013638	0.106934	0.031806	7.840763444	2.332125942
Glass Scene	0.009809	0.075703	0.033609	7.717601843	3.426273294
Non-Glass Scene	0.013581	0.092971	0.026781	6.845536477	1.971917062
Changing Lighting Scene	0.006818	0.035554	0.023531	5.214722577	3.451254748
Stable Lighting Scene	0.008489	0.040613	0.028436	4.784235885	3.349786191

**Table 9 sensors-24-07619-t009:** APE assessment.

APE	FIS-IESKF Compared to Ground Truth		EKF Compared to Ground Truth		UKF Compared to Ground Truth	
	Translation (m)	Rotation (°)	Translation (m)	Rotation (°)	Translation (m)	Rotation (°)
RMSE_APE_	0.009396	0.149480	0.054120	64.39633	0.016228	81.32271
max_APE_	0.024468	1.956488	0.241905	180.0000	0.060894	179.9711
min_APE_	0.000000	0.003514	0.000000	0.022666	0.000957	0.014259
mean_APE_	0.008042	0.028733	0.045870	49.96441	0.014387	60.38921

**Table 10 sensors-24-07619-t010:** RPE assessment.

RPE	FIS-IESKF Compared to Ground Truth		EKF Compared to Ground Truth		UKF Compared to Ground Truth	
	Translation (m)	Rotation (°)	Translation (m)	Rotation (°)	Translation (m)	Rotation (°)
RMSE_RPE_	0.002347	0.201990	0.005863	14.87776	0.004908	12.363485
max_RPE_	0.011857	1.950079	0.026973	179.9996	0.017201	180.00000
min_RPE_	0.000000	0.000000	0.000000	0.000000	0.000000	0.000000
mean_RPE_	0.001761	0.042539	0.004892	1.647146	0.004165	1.440816

**Table 11 sensors-24-07619-t011:** Computational Complexity of FIS-IESKF method.

Name of Steps	IESKF Initialization	IESKF Prediction with WO-FIS		IESKF Observation with VO-FIS		IESKF Error Compensation	IESKF Error Reset
Steps	①	②	⑥	③	⑦	④	⑤
Asymptotic Computational Complexity	*O*(54)	*O*(5832)	*O*(46)	*O*(999)	*O*(20)	*O*(18)	*O*(18)
Specific Computational Complexity	1·*O*(54)	*Ts*·*O*(5832)	*Ts*·*O*(46)	*Ts*·*O*(999)	*Ts*·*O*(20)	*Ts*·*O*(18)	*Ts*·*O*(18)
Time Proportion	0.00039%	84.11%	0.66%	14.41%	0.29%	0.26%	0.26%

## Data Availability

The proposed method with codes can be found in our GitHub repository (https://github.com/botlowhao/FIS-IESKF, accessed on 7 November 2024). The experimental images and the codes of the proposed multi-fusion method can be found in this publicly shared repository.

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
