# Peer review of "Multi-Sensor Fusion for Wheel-Inertial-Visual Systems Using a Fuzzification-Assisted Iterated Error State Kalman Filter"

_sensors, 2024, doi:10.3390/s24237619_

Round 1

Reviewer 1 Report

Comments and Suggestions for Authors

Attached

Reviewer 2 Report

Comments and Suggestions for Authors

(1)  For introduction section, Literature review should be more detailed and comprehensive. Authors should add more recent research progress to this part and give a brief introduction of the development history on your topic. The novelty of  multi-fusion method using a fuzzy inference system to optimize the process and observe noise covariance in the Iterative Error State Kalman Filter should be highlighted.

(2) It is better for authors to clarify your objectives of your study in the Introduction section.

(3) For your proposed method, advantages compared to other methods should be clarified. The evaluations are not comprehensive.

(4) Inertial odometry information is derived from the fusion of data from the accelerometer, gyroscope, and magnetometer in a 9-axis IMU. The advantage of an IMU-based navigation system is that it is entirely independent, relying on no external signals, allowing it to function reliably even when GPS is unavailable. Could you give some examples to justify this?

(5) Reasons why you choose this method should be added to your manuscript.

(6) In the comparative study, authors can use other methods to analyze the same problem and highlight your method's accuracy.

(7) The mathematical presentation of this paper needs to be improved. For example, X-axis, here X is a variable, it should be in italics form. There are many similar issues. 

(8) For conclusions, more detailed results should be presented.

(9) Finally, the language of the paper is poor. It must be improved in the revised version. I recommend a professional editing service if possible. The language sometimes makes the paper hard to understand. 

Comments on the Quality of English Language

The manuscript's language quality needs significant enhancement to meet the publication standards. It currently lacks clarity, which affects the reader’s ability to fully understand the content. A thorough review by a professional language editing service is highly recommended. If substantial improvements cannot be made, the paper may not be suitable for acceptance.

Reviewer 3 Report

Comments and Suggestions for Authors

The proposed multi-sensor fusion framework using a Fuzzy Inference System (FIS) with Iterative Error State Kalman Filter (IESKF) presents a valuable approach to enhancing the accuracy and robustness of pose estimation for indoor mobile robots. The integration of FIS to dynamically adjust noise covariance in response to variable kinematic and environmental factors is innovative and shows promise in addressing the challenges of odometry drift and sensor noise in indoor environments. This method has potential applications in autonomous navigation and inspection tasks, especially in GPS-denied spaces, making it relevant and timely for the journal's readership.

1. The paper does not provide sufficient detail on the rationale behind the selection of fuzzy rules and membership functions for the FIS. How do the authors define these parameters?

2. Integrating FIS with IESKF can introduce computational overhead, which may impact real-time performance, especially on low-power or resource-limited platforms. A computational complexity analysis should be given to demontrate the method's scalability and feasibility for various robotic applications.

3. While the paper includes comparisons with EKF, it would benefit from a more extensive comparison with other state-of-the-art sensor fusion and SLAM algorithms used for indoor localization.

4. The paper's experiment is limited  to relatively controlled indoor environments. how do authors address the choice of dynamic and changing environments, such as dynamic obstacles, irregular lighting, and varying surface conditions.

Comments on the Quality of English Language

The manuscript can benefit from language editing.

Round 2

Reviewer 1 Report

Comments and Suggestions for Authors

The authors have done a good job of correcting the errors. The reviewer recommends eliminating two more inaccuracies, after which the article can be published.

1. In formula (11), specify the argument of the integrable function under the integral and replace the differential dt with ds, for example. This will emphasize that the integral and dt in the equation itself use different time variables.

2. The authors' arguments about the nature of covariances in the EKF equations are unnecessary and incorrect. It's much simpler. For example. in (27), the value of P_t is not "the posterior estimate covariance." !!! This is only an empirical assessment, which in practice may turn out to be arbitrarily bad. As in the first review, the reviewer points out the need to exclude such incorrect statements from the text. This can be done with the word "approximation".

Reviewer 2 Report

Comments and Suggestions for Authors

The paper has been improved. I recommend it for publication. 
